# Retinoblastoma protein as an intrinsic BRD4 inhibitor modulates small molecule BET inhibitor sensitivity in cancer

Donglin Ding[1,2], Rongbin Zheng[3,4], Ye Tian[5], Rafael Jimenez [6], Xiaonan Hou[7], Saravut J. Weroha[7], Liguo Wang [8], Lei Shi [1,9] ✉ & Haojie Huang [1,2,10] ✉

Bromodomain and extraterminal (BET) proteins including BRD4 play important roles in oncogenesis and immune inflammation. Here we demonstrate that cancer cells with loss of the retinoblastoma (RB) tumor suppressor became resistant to small molecule bromodomain inhibitors of BET proteins. We find that RB binds to bromodomain-1 (BD1) of BRD4, but binding is impeded by CDK4/6-mediated RB phosphorylation at serine-249/threonine-252 (S249/T252). ChIP-seq analysis shows RB knockdown increases BRD4 occupancy at genomic loci of genes enriched in cancer-related pathways including the GPCR-GNBIL-CREB axis. S249/T252-phosphorylated RB positively correlates with GNBIL protein level in prostate cancer patient samples. BET inhibitor resistance in RB-deficient cells is abolished by co-administration of CREB inhibitor. Our study identifies RB protein as a bona fide intrinsic inhibitor of BRD4 and demonstrates that RB inactivation confers resistance to small molecule BET inhibitors, thereby revealing a regulatory hub that converges RB upstream signaling onto BRD4 functions in diseases such as cancer.

Bromodomain and extraterminal (BET) proteins, including BRD2, BRD3, BRD4, and BRDT (a testis-specific member), are important factors to recognize acetylated histones and play a critical role in transcriptional activation[1]. There are two bromodomains in BET proteins[2] and it has been shown that the activity of bromodomain-1 (BD1) is primarily related to cancer whereas bromodomain-2 (BD2) appears to mainly function in immune response and inflammation[3]. A few small molecule inhibitors, especially those targeting BD1 domain have been developed and manifested some promising anti-cancer activities in a series of clinical trials[4–6]. However, resistance to BET inhibitors often

occurs and the underlying mechanisms are not fully understood[7]. Thus, there is an urgent need to develop new strategies or compounds to overcome BET inhibitor resistance. Additionally, it has been reported that the function of BRD4 can be modulated by other factors through protein posttranslational modifications such as phosphorylation, methylation, and ubiquitylation[8–10]. However, it remains unclear whether there exists any endogenous protein(s) that functions as intrinsic inhibitors of BRD4.

Retinoblastoma (RB) protein, encoded by *RB1* gene acts as a tumor suppressor by interacting with others proteins such as E2F

[1]Department of Biochemistry and Molecular Biology, Mayo Clinic College of Medicine and Science, Rochester, MN 55905, USA. [2]Department of Urology, Mayo Clinic College of Medicine and Science, Rochester, MN 55905, USA. [3]Basic and Translational Research Division, Department of Cardiology, Boston Children's Hospital, Boston, MA 02115, USA. [4]Department of Pediatrics, Harvard Medical School, Boston, MA 02115, USA. [5]Department of Urology, Jiangsu Province Hospital of Chinese Medicine, Affiliated Hospital of Nanjing University of Chinese Medicine, Nanjing 210029, China. [6]Department of Laboratory Medicine and Pathology, Mayo Clinic College of Medicine and Science, Rochester, MN 55905, USA. [7]Divison of Oncology, Mayo Clinic College of Medicine and Science, Rochester, MN 55905, USA. [8]Divison of Medical Informatics and Statistics, Mayo Clinic College of Medicine and Science, Rochester, MN 55905, USA. [9]Department of Radiation Oncology, Cancer Center, Zhejiang Provincial People's Hospital, Affiliated People's Hospital of Hangzhou Medical College, Hangzhou 310000, China. [10]Mayo Clinic Cancer Center, Mayo Clinic College of Medicine and Science, Rochester, MN 55905, USA. ✉e-mail: shileihmu@gmail.com; huang.haojie@mayo.edu

family proteins to inhibit cancer-promoting activities such as accelerated cell cycle progression[11]. *RB1* gene is frequently deleted in human cancers including prostate cancer (PCa)[12–14]. In addition to the deletion or mutation in the *RB1* gene, the cell cycle-inhibitory function of RB protein is impeded by phosphorylation of RB protein mediated by Cyclin-dependent kinases (CDKs) such as CDK4/6[15–18]. Studies from us and others have showed that the RB amino-terminal region (RB-N) regulates cancer cell growth and immune response by interacting with an FXXXV motif in client proteins including EID-1 and p65, respectively. Notably, phosphorylation of serine 249/threonine 252 (S249/T252) in the linker region of RB-N by CDK4/6 either enhances or inhibits the binding of its partners and the effect is dependent on the charge composition surrounding the FXXXV motif[19,20]. However, it remains largely unclear whether RB-N could also regulate other cancer-relevant functions through undefined mechanisms.

In our current study, we find that RB loss leads to BET inhibitor resistance in a manner independent of the RB-E2F1 pathway in PCa cells. We show that RB-N binds to the BD1 domain of BRD4 and restricts BRD4 occupancy in the genome; however, RB-N-BRD4 binding is attenuated by RB-N phosphorylation at S249/T252 residues. Our data also reveal that depletion of RB enhances BRD4 occupancy at the loci of cancer-related genes including the GPCR-cAMP signaling gene *GNB1L*. Finally, we demonstrate that loss of RB protein confers resistance to small molecule BET inhibitor and this effect is abolished by co-inhibition of GNB1L-CREB signaling.

## Results

### RB deficient cells are resistant to small molecule BET inhibitors
The *RB1* gene is frequently deleted in many human cancers including neuroendocrine PCa (NEPC), an advanced form of PCa which is resistant to most existing treatments in clinic[21]. We, therefore, sought to investigate how RB loss affects drug resistance in PCa cells with NEPC traits. It has been reported previously that PC-3 PCa cells exhibit certain characteristics of NEPC[22]. We generated control or RB knockdown (KD) PC-3 PCa cells and treated them with a group of drugs or chemicals that are used in clinical or pre-clinical settings. As expected, PC-3 cells became resistant to the CDK4/6 inhibitor palbociclib when RB was knocked down (Fig. 1a and Supplementary Fig. 1a). Consistent with our previous finding[19], RB-deficient cells were resistant to the NF-κB inhibitor JSH23 (Fig. 1a). Notably, these cells were also resistant to bromodomain inhibitors including BET BD1 inhibitors JQ1 and I-BET726 as well as CBP/p300 inhibitor CPI-637 (Fig. 1a and Supplementary Fig. 1b). RB knockdown also caused BET BD1 inhibitor resistance in another aggressive PCa cell line C4-2 (Fig. 1b and Supplementary Fig. 1a). Additionally, RB knockdown conferred resistance to BET inhibitors in LNCaP cells, a hormone-sensitive PCa cell line (Supplementary Fig. 1a, c). These results suggest that resistance to BET inhibitors could be a common phenomenon in RB-deficient PCa cells regardless of neuroendocrine phenotype. Further analysis showed that treatment of both PC-3 and C4-2 parental cells with JQ1 induced apoptotic cell death as evident by the cleavage of PARP (c-PARP) and Caspase-3 (c-Caspase-3), but these effects were abolished by RB KD (Fig. 1c, d). The inhibition of JQ1-induced apoptosis by RB KD was unlikely mediated through RB KD-induced upregulation of NF-κB target genes because expression of these genes was equivalently blocked by JQ1 in both control and RB KD cells (Supplementary Fig. 1d, e). These data suggest that RB loss blocks BET inhibitor-induced cell death through mechanisms other than the NF-κB signaling.

Due to the pivotal role of RB in regulating E2F1 activity, we examined the contribution of aberrantly activated E2F1 to BET BD1 inhibitor resistance in RB-deficient cells. It has been shown recently that RB deficiency induces JQ1 resistance in NUT midline carcinoma cells and such resistance is mediated by cell cycle regulators including E2F1[23]. In PC-3 cells, however, we found that while JQ1 treatment

downregulated expression of c-Myc, a known JQ1 inhibitory target[24], it had no obvious effect on E2F1 expression (Supplementary Fig. 1f). DU145 is a PCa cell line in which one allele and exon 21 in the other allele of the *RB1* gene are deleted[25]. To further test our hypothesis, we transfected DU145 cells with either wild-type (WT) RB or the E2F1-binding deficient mutant R661W[26]. As expected, reciprocal co-immunoprecipitation (co-IP) assays showed that R661W mutant failed to bind to E2F1 (Supplementary Fig. 1g). However, we demonstrated that increased expression of both WT RB and R661W equivalently sensitized DU145 cells to BET BD1 inhibitors JQ1 and iBET (Fig. 1e, f). Furthermore, E2F1 KD failed to reverse RB loss-mediated BET inhibitor resistance in PC-3, C4-2, and LNCaP cell lines (Fig. 1g, h and Supplementary Fig. 2a–d). Together, our data suggest that RB loss induces BET inhibitor resistance in a manner independent of E2F1 signaling in PCa cell lines examined.

### RB-N interacts with BRD4 and the interaction is diminished by RB phosphorylation
It is known that RB binds to and regulates the activity of its target proteins, such as E2F1 and p65. We hypothesized that RB modulates BET inhibitor sensitivity by binding to and regulating BET protein activity. To test this hypothesis, we first performed co-IP experiments. We demonstrated that the ectopically expressed HA-RB interacted with BRD2, BRD3, and BRD4 in 293T cells (Supplementary Fig. 3a). RB interaction with these BET proteins were confirmed at the endogenous level in both C4-2 and PC-3 cells (Fig. 2a, b). RB protein is highly phosphorylated in proliferating cells. We sought to determine whether RB phosphorylation affects RB-BET protein interaction. 293T cells were transfected with HA-RB and cell lysate was treated with λ protein phosphatase prior to co-IP with anti-HA antibody. We demonstrated that phosphatase treatment largely enhanced RB interaction with BRD4, but not BRD2 and BRD3 (Supplementary Fig. 3b). Similar results were obtained at the endogenous level in both C4-2 and PC-3 cell lines (Fig. 2c, d). Moreover, we found that RBΔCDK mutant, in which fifteen CDK serine/threonine phosphorylation residues were mutated to alanine[16], had greater interaction with BRD4 than the WT counterpart (Supplementary Fig. 3c). These data indicate that RB phosphorylation impairs RB-BRD4 interaction.

To define which region in RB mediates its interaction with BET proteins, we constructed three glutathione-S-transferase (GST)-RB recombinant protein vectors (Supplementary Fig. 3d) and purified the recombinant proteins from bacteria. Results from the GST pulldown assay showed that BRD4 specifically bound to the N-terminal portion of RB (RB-N) in vitro (Fig. 2e). In contrast, BRD2 and BRD3 interacted with both N- and C-terminal parts of RB, but the interactions were much weaker compared to the BRD4-RB-N interaction (Fig. 2e). Therefore, we focused on RB regulation of BRD4 in further studies. Co-IP assay not only confirmed that BRD4 interacted with RB-N in cells, but also demonstrated that their interaction is regulated by phosphorylation (Fig. 2f).

### RB serine-249/threonine 252 phosphorylation disrupts RB-N-BRD4 interaction
Since dephosphorylation of RB-N enhances its interaction with BRD4, we decided to determine which phosphorylation sites are involved in this process. GST pulldown assay revealed that BRD4 bound to the N-terminal portion of RB-N which contains CDK4/6 phosphorylation sites serine-249 and threonine-252 (S249/T252), but no interaction between BRD4 and the C-terminal portion of RB-N was detected (Fig. 2g, h). We also found that treatment of CDK4/6 inhibitor palbociclib or co-knockdown of CDK4 and CDK6 enhanced RB-N interaction with BRD4 (Supplementary Fig. 3e, f). Moreover, the BRD4-RB-N interaction was largely enhanced by the S249/T252 phosphorylation-resistant mutant S249A/T252A or deletion of the S249/T252-containing motif [249]SPRT[252] (RB-NΔSPRT) (Fig. 2i and Supplementary Fig. 3g).

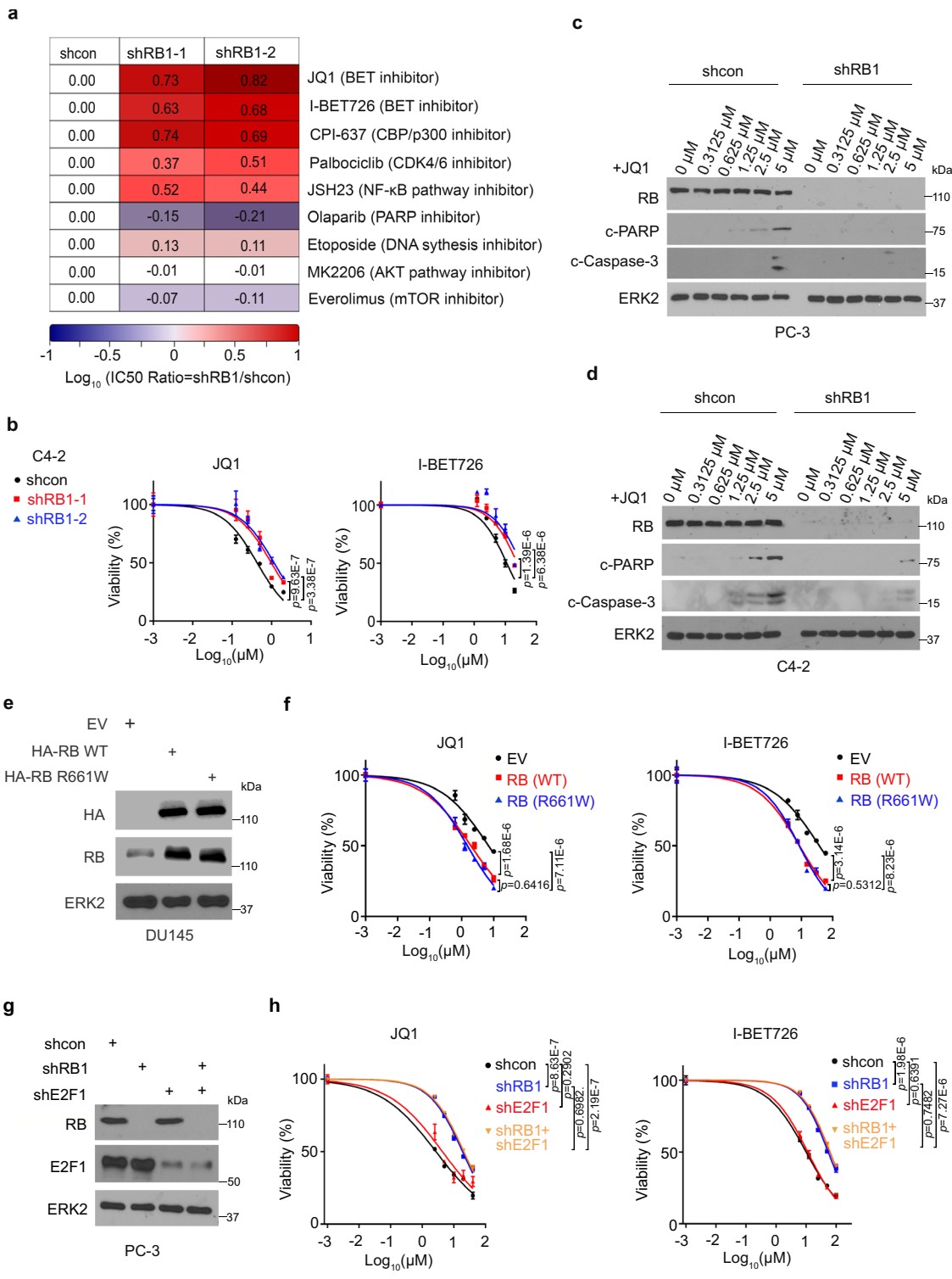

**Fig. 1 | RB loss confers resistance to BET inhibitors independent of E2F1 signaling in PCa cells. a** Heatmap showing the effects of a set of small molecule inhibitors of cancer-related pathways on growth of control and RB knockdown PC-3 cells. **b** Cell viability analysis of control and RB knockdown C4-2 cells after the treatment with different concentrations of JQ1 or i-BET726 for 72 h. Western blot (WB) analysis of cleaved-PARP (c-PARP) and cleaved caspase-3 (c-Caspase-3) level in control and RB knockdown PC-3 (**c**) and C4-2 cells (**d**) treated with different doses of JQ1 for 72 h. WB analysis of indicated proteins (**e**) in DU145 cells after transfected with empty vector (EV), HA-RB or mutant HA-RB R661W for 72 h followed by cell viability analysis (**f**) after treatment with different doses of BET inhibitors for 72 h. WB analysis of indicated proteins (**g**) in PC-3 cells after infected with lentivirus for control shRNA or gene-specific shRNAs for RB1 or E2F1 for 48 h followed by cell viability analysis (**h**) after treatment with different doses of BET inhibitors for 72 h. Data in **a**, **b**, **f**, and **h** represented as mean ± s.d. from triplicates. Statistical significance in **b**, **f**, and **h** was assessed by two-way ANOVA.

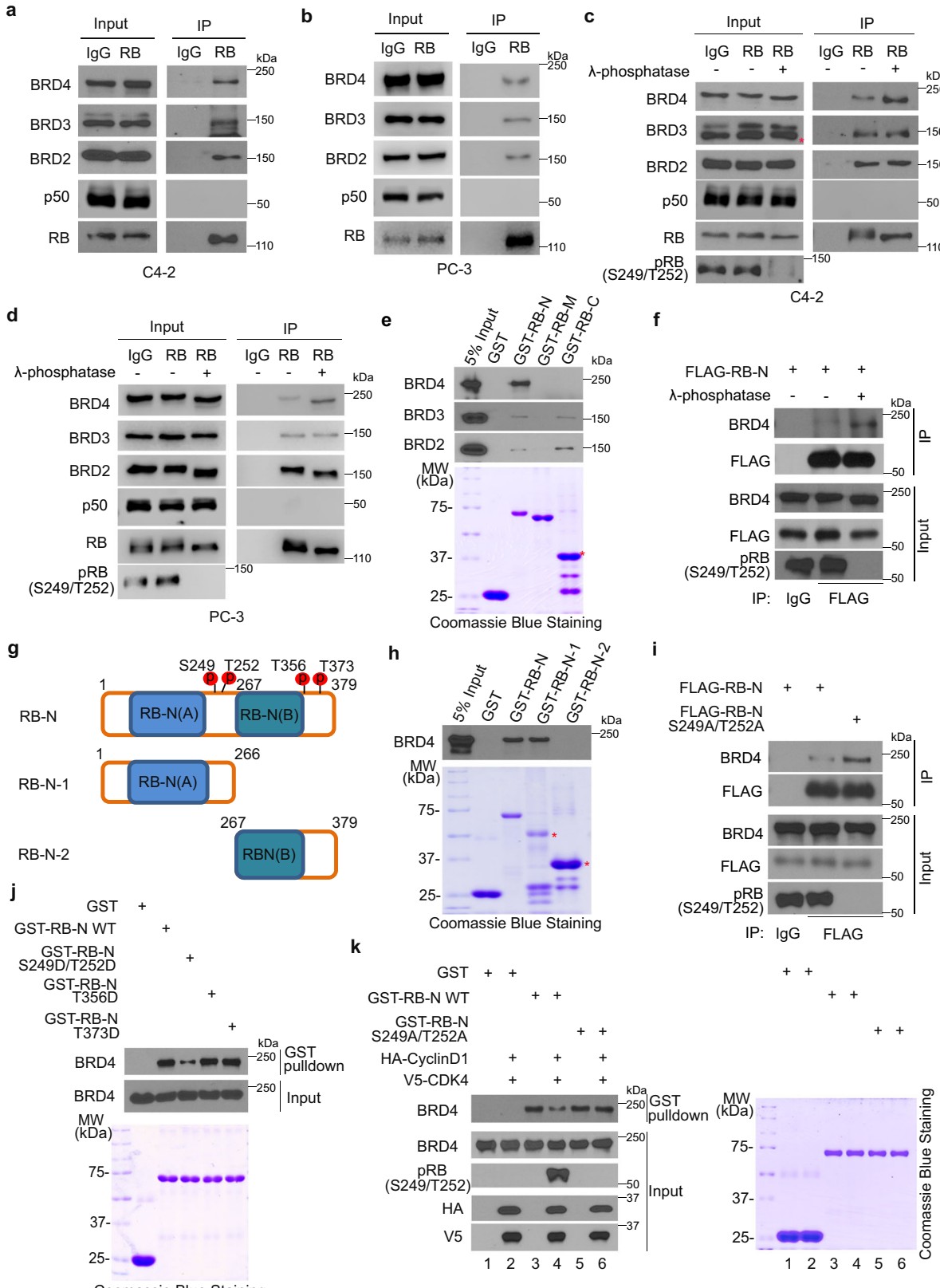

These results suggest that S249/T252 phosphorylation inhibits BRD4-RB-N interaction. To mimic the RB-N phosphorylation at S249 and T252, we mutated S249 and T252 to aspartic acid (D). By performing GST pulldown assay, we demonstrated that the S249D/T252D mutant had less binding ability to BRD4 compared to the WT counterpart (Fig. 2j). In contrast, phosphorylation-mimicking mutation T356D/

T373D had no effect on BRD4-RB-N binding (Fig. 2j), consistent with the observation that the C-terminal portion of RB-N does not interact with BRD4 (Fig. 2h). Next, we performed in vitro kinase assay prior to GST pulldown assay. We demonstrated that CDK4/6 diminished RB-N-BRD4 interaction; however, this effect was abolished by S249A/T252A or RB-NΔSPRT mutant (Fig. 2k and Supplementary Fig. 3h).

**Fig. 2 | The dephosphorylation of RB-N (S249/T252) enhances the interaction between RB-N and BRD4.** WB analysis of indicated proteins in input and IP samples from C4-2 (**a**) and PC-3 cells (**b**). The NFκB protein p50 was used as a negative control. WB analysis of indicated proteins in input and IP samples from C4-2 (**c**) and PC-3 cell lysate (**d**) treated with or without λ protein phosphatase. **e** Top, WB analysis of BET proteins from 293T cell lysate pulled down by GST-tagged truncated RB recombinant proteins. Bottom, Coomassie blue staining of GST and GST-RB-N recombinant proteins. **f** WB analysis of indicated proteins in input and IP samples from 293T cell lysate treated with or without λ protein phosphatase. **g** Diagram showing four CDK-phosphorylation sites in RB-N. **h** Top, WB analysis of BRD4 from 293T cell lysate pulled down by GST-RB-N recombinant proteins. Bottom, Coomassie blue staining of GST and GST-RB-N recombinant proteins. **i** WB analysis of indicated proteins in input and IP samples from 293T cells transfected with indicated plasmids. **j** WB analysis of BRD4 protein in 293T cell lysate pulled by the recombinant proteins of GST-RB-N and indicated mutants. **k** WB analysis of BRD4 proteins in 293T cell lysate pulled down by the recombinant proteins of GST-RB-N and indicated mutants phosphorylated through in vitro kinase assay. All the western blot assays were repeated two times independently with similar results.

Collectively, these data suggest that S249/T252 phosphorylation by CDK4/6 impedes RB binding of BRD4.

## The [157]FLQKI[161] motif in BRD4 BD1 domain mediates BRD4-RB-N interaction

To determine the region in BRD4 that is required for RB binding, we generated four FLAG-tagged truncation mutants of BRD4 for co-IP assay. We found that only the BRD4 N-terminal region which contains two bromodomains (amino acids 1 to 470) bound to RB (Fig. 3a, b). Both GST pulldown and co-IP assays further showed that bromodomain 1 (BD1), but not BD2 binds RB (Fig. 3c, d and Supplementary Fig. 4a).

We and others have previously shown that an FXXXV motif in the RB-binding partners is important for their binding with RB[19,20], we sought to determine whether there is any functional FXXXV motif in BRD4 that mediates binding to RB. We noticed that there is one typical FXXXV motif ([83]FQQPV[87]) and an atypical FXXXI motif ([157]FLQKI[161]) in BD1, the RB-binding region in BRD4 (Fig. 3e). To test the functionality of these two FXXXV/I motifs, we generated three alanine mutants (Fig. 3f) and utilized them for in vitro GST pulldown assay. While the [83]AQQPA[87] mutant alone had no effect on BRD4 BD1 binding of RB, the [157]ALQKA[161] mutation alone largely diminished and [83]AQQPA[87]/[157]ALQKA[161] double mutant almost completely lost the binding of RB (Fig. 3g). Similar results were obtained in co-IP assay performed in cell lysate (Fig. 3h). Given that RB interacts with BRD4 through BD1, a domain critical for BRD4 recognition of acetylated histone and non-histone proteins[27,28], we examined whether inhibition of the bromodomain activity by JQ1 affects the RB-BRD4 interaction. We found that administration of JQ1 to the GST pulldown and co-IP assay buffers had no impact on BRD4 binding of RB (Supplementary Fig. 4b–e).

Among the three members of the pocket protein family, we found that only RB, but not other two related proteins p107 and p130 can bind BRD4 (Fig. 3i), implicating that there is a unique motif in RB for its preferential binding of BRD4. In agreement with this observation, RB protein is the only member of the pocket protein family that harbors an R-linker which contains BRD4-interacting sites S249/T252. Notably, there are a number of positively charged arginine (R) residues within the R-linker in RB-N and several negatively charged glutamic acid (E) residues surrounding the [157]FLQKI[161] motif in BRD4 (Fig. 3j, top). As described above, phosphorylation of RB-N S249/T252 impedes the BRD4-RB-N interaction. We hypothesized that the negative charges in the proximity of [157]FLQKI[161] are critical for BRD4 interaction with RB R-linker (Supplementary Fig. 4f, left panel). To test this hypothesis, we made an E-to-A mutant by replacing E residues with alanine (E/A mutants) surrounding [157]FLQKI[161] and an E-to-R mutant by replacing E residues with positively charged arginine (E/R mutant) (Fig. 3j, bottom) and performed GST pulldown assay. We demonstrated that the interaction between BRD4 BD1 and RB was attenuated by the E/A mutant and the interaction was further reduced by the E/R mutant (Fig. 3k). Together, we provide evidence that the [157]FLQKI[161] motif in BD1 and the negative charge in the adjacent residues are important for BRD4 binding of RB and that this interaction is diminished by the addition of negative charges in the RB R-linker due to S249/T252 phosphorylation by CDK4/6 (Supplementary Fig. 4f, right). To investigate the effect of RB on the binding ability of BRD4 to chromatin, we carried out the chromatin-binding assay and found that RB loss increased BRD4 binding of chromatin while the restoration of RB in RB-deficient cells reduced BRD4 binding on chromatin (Fig. 3l, m).

## RB loss activates the GNB1L-CREB signaling axis through BRD4

BD1 is critical for BRD4 to recognize the acetylation moiety on histones and participates in transcription control of cancer-related genes[2,3]. Our data support the notion that RB binds to BD1 and limits BRD4 activity in chromatin binding and that RB loss renders resistance to BET inhibitors by increasing BRD4 engagement with chromatin and inducing transcriptional reprogramming. To interrogate this hypothesis, we performed BRD4 ChIP-seq in control and RB KD C4-2 cells. By running the Cistrome pipeline, we revealed that the quality of the BRD4 ChIP-seq data obtained from both control and RB KD C4-2 cells were very comparable to the BRD4 ChIP-seq results of approximately 300 different human cell samples in the Cistrome database (Supplementary Fig. 5a−l and Supplementary Data 1). We demonstrated that RB knockdown increased BRD4 occupancy in the genome ($n = 107$, $Log_2$ (fold change) $> 1$, $p < 0.05$), but only reduced BRD4 binding at 4 loci (Fig. 4a, b, Supplementary Data 2 and 3). The 107 peaks with enhanced occupancy of BRD4 were related to 152 genes, some of which were enriched in the cAMP pathway (Fig. 4c). cAMP production functions downstream of the GPCR signaling. We demonstrated that RB depletion increased BRD4 occupancy at certain region(s) in the loci of genes in the GPCR-cAMP pathways such as *GNB1L* (Fig. 4d and Supplementary Fig. 6a, b). Using ChIP-qPCR assay, we confirmed the increased BRD4 occupancy in the loci of GPCR-cAMP genes such as *EDNRA*, *GNB1L*, *GRIN3A*, *GRM4*, and *SSTR1* in both PC-3 and C4-2 cells (Fig. 4e, f). Consistent with the ChIP-qPCR data, expression of these five genes was upregulated after RB KD in both PC-3 and C4-2 cells except *GRIN3A* and *GRM4* expression in PC-3 cells (Fig. 4g, h). GNB1L protein was also substantially upregulated in RB KD PC-3 and C4-2 cells (Fig. 4i). We further showed that knockdown of BRD4 alone not only decreased GNB1L expression and phosphorylation of CREB, a downstream effector of the GPCR-cAMP pathway, but also completely abolished RB KD-induced upregulation of GNB1L protein and CREB phosphorylation (Fig. 4j, k), suggesting that RB deficiency-induced upregulation of GNB1L is mediated through BRD4. Notably, similar results were obtained in PCa patient samples from the TCGA database[29]. We found that *GNB1L* mRNA expression was much higher in RB deletion samples compared to RB WT samples (Fig. 4l, m), and their expression was inversely correlated (Fig. 4n). Among other four GPCR-cAMP pathway genes expression of two of them (e.g., *GRIN3A* and *GRM4*) was also upregulated in RB-deficient tumors compared to Rb WT counterparts in the TCGA cohort and the difference was statistically significant (Supplementary Fig. 6c−f). These data suggest that RB can regulate expression of these GPCR-cAMP genes in cultured PCa cells and patient samples, but the regulation at certain gene loci could be influenced by the cellular contexts, especially in PCa samples from patients.

The high-level expression of *GNB1L* mRNA also significantly associated with poor survival of PCa patients (Fig. 4o). Accordingly, we showed that ectopic overexpression of GNB1L caused resistance to

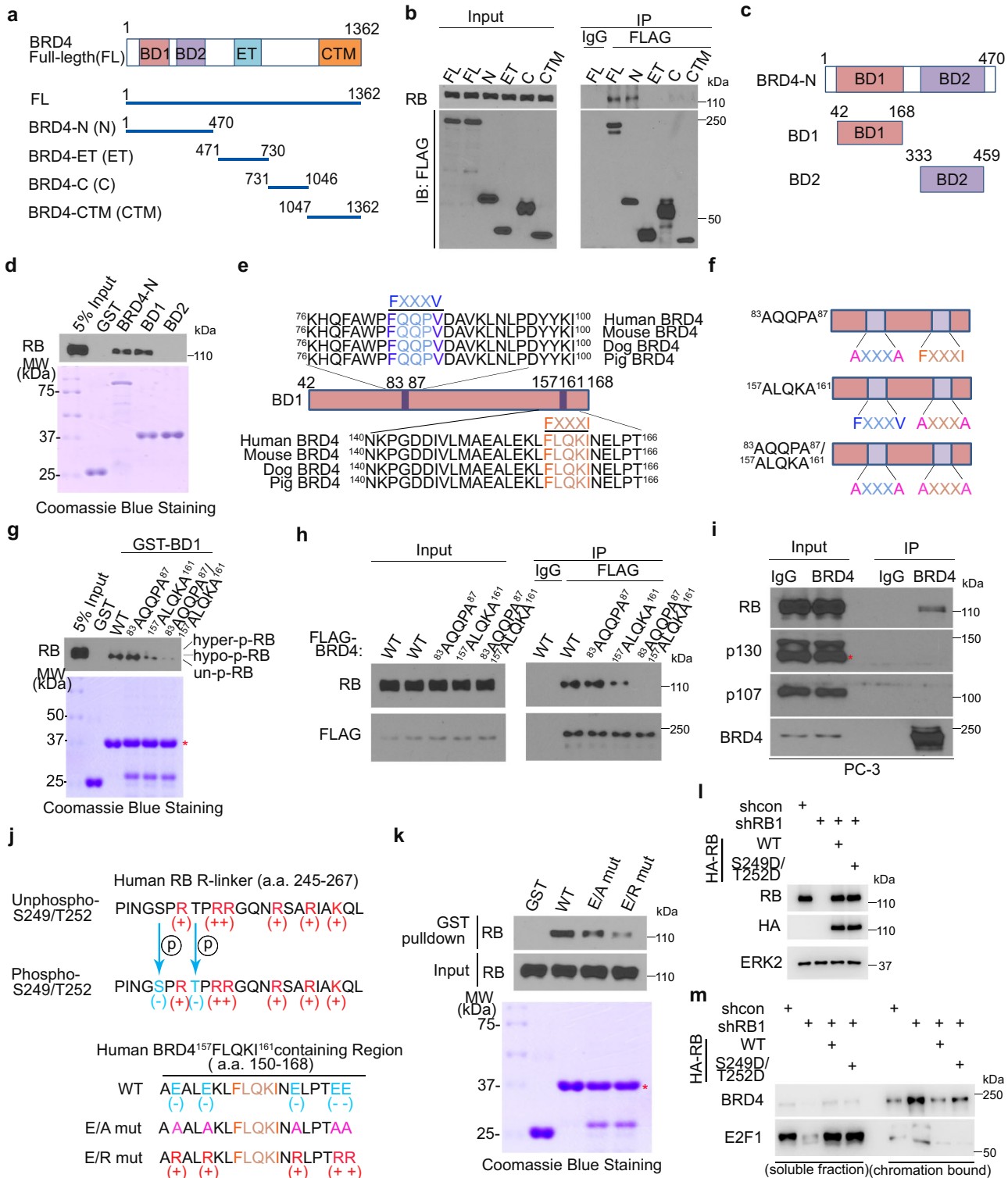

**Fig. 3 | BRD4 interacts with RB through an FXXXI motif in the BD1 domain.**
**a** Diagram showing different mammalian expression vectors for full-length (FL) and truncated BRD4 proteins. **b** WB analysis of RB protein in input and co-IP samples from C4-2 cells transfected with different fragments of BRD4. **c** Diagram showing GST-BRD4 recombinant proteins containing two or each single BD domain. **d** Top, WB analysis of RB proteins from C4-2 cell lysate pulled down by different GST-RB-N recombinant proteins. Bottom, Coomassie blue staining of GST and GST-RB-N recombinant proteins. **e** Diagram showing the location of FXXXV and FXXXI motifs in BD1 of BRD4. **f** Diagram showing single or double mutants of FXXXV and FXXXI motifs. **g** Top, WB analysis of RB proteins from C4-2 cell lysate pulled down by different GST-BD1 and mutant recombinant proteins. Bottom, Coomassie blue staining of GST and GST-BD1 and mutant recombinant proteins. **h** WB analysis of RB

proteins from the co-IP samples from C4-2 cells expressing indicated FLAG-tagged BRD4 WT or mutants. **i** WB analysis of indicated protein in input and co-IP samples from PC-3 cells. **j** Diagram showing the charge composition in the RB-N R-linker region surrounding S249/T252 and residues surrounding the FXXXI motif in BRD4 BD1. **k** Top, WB analysis of RB proteins from C4-2 cell lysate pulled down by different GST-BRD4 BD1 and mutants. Bottom, Coomassie blue staining of GST and GST-BRD4 BD1 recombinant proteins. **l** WB analysis of indicated proteins in C4-2 cells transfected with indicated plasmids. **m** WB analysis of soluble and chromatin fractions isolated from PC-3 cells expressing the indicated proteins. Western blot assays in **b**, **d**, **g**, **h**, **i**, **k**, **l**, and **m** were repeated two times independently with similar results.

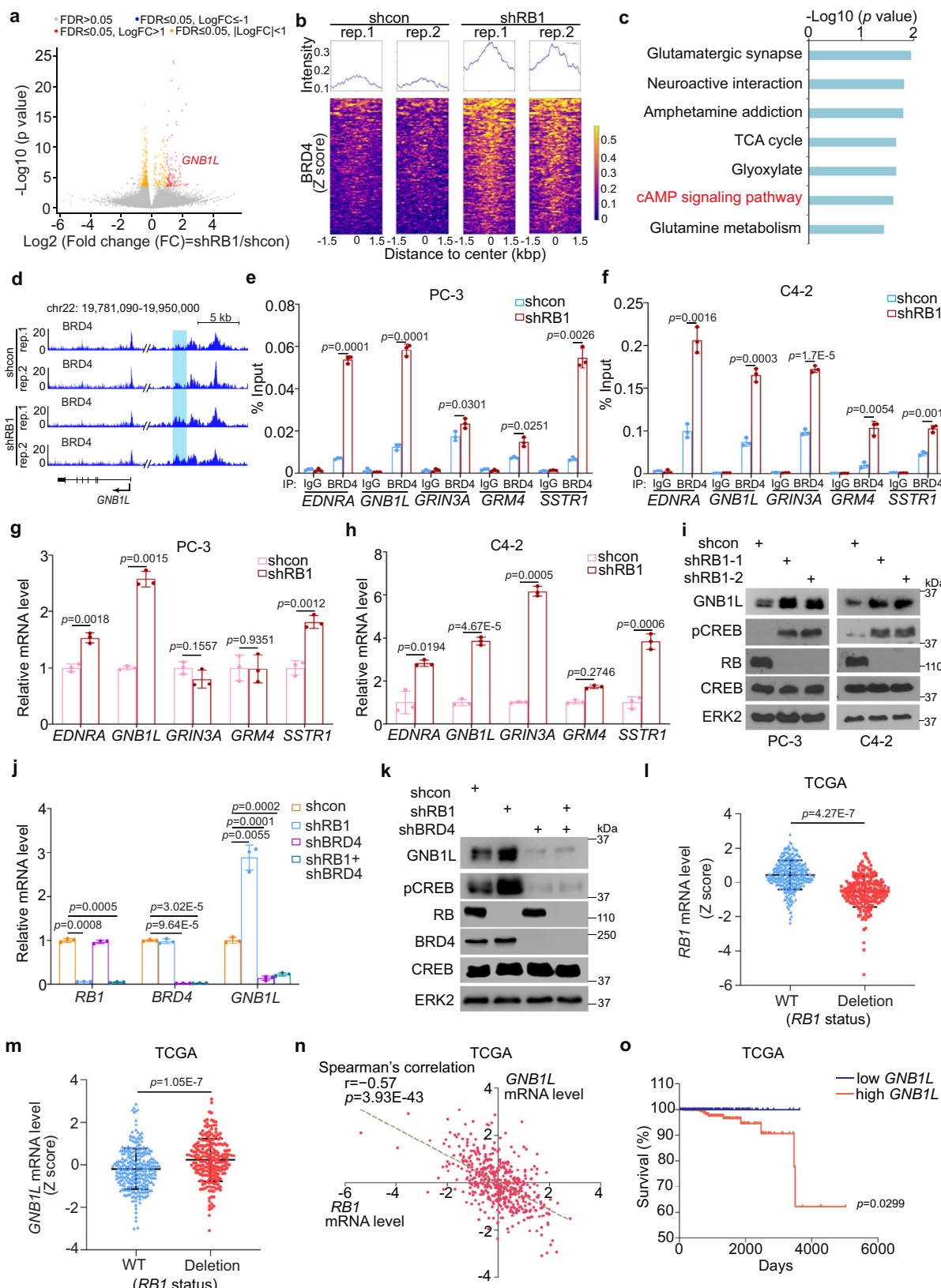

BET inhibitors in both PC-3 and C4-2 cells (Supplementary Fig. 7a–d). Moreover, knockout of GNB1L re-sensitized RB-deficient PC-3 and C4-2 cells to BET inhibitors (Supplementary Fig. 7e–h). Similar to the findings in LNCaP/AR cell line[30], RB KD not only decreased AR mRNA expression, but also induced resistance to the antiandrogen enzalutamide in C4-2 cells (Supplementary Fig. 7i, j). However, co-knockdown

of GNB1L failed to abolish RB KD-induced AR downregulation and enzalutamide resistance (Supplementary Fig. 7i, j). Moreover, similar to the findings in LNCaP/AR cell line[30], we found that RB KD in C4-2 cells induced upregulation of neuron-specific enolase (NSE), a neuroendocrine cell marker (Supplementary Fig. 7i); however, this effect was not affected by co-knockdown of GNB1L (Supplementary Fig. 7i).

**Fig. 4 | cAMP pathway gene GNB1L is a BRD4 binding target gene whose expression is upregulated upon RB loss. a** Volcano plot showing BRD4 ChIP-seq peaks up- (red) or down-regulated (blue) in C4-2 cells expressing control (shcon) or RB-specific shRNAs. **b** Heatmaps show the signaling intensity of 107 enhanced BRD4 binding peaks in C4-2 cells due to RB KD. **c** KEGG pathway analysis of 152 genes related to 107 genome loci with increased BRD4 occupancy upon RB loss. **d** UCSC Genome Browser screenshots showing the enhanced BRD4 occupancy in the *GNB1L* gene locus in shRB C4-2 cells compared to shcon cells. ChIP-qPCR analysis of BRD4 occupancy at *GNB1L* and other GPCR-cAMP gene loci after RB loss in PC-3 (**e**) and C4-2 cells (**f**). RT-qPCR analysis of the mRNA level of the indicated GPCR-cAMP pathway genes in control or RB knockdown PC-3 (**g**) and C4-2 (**h**) cells. **i** WB analysis of expression of GNB1L and phosphorylation (activation) of CREB after RB knockdown in PC-3 and C4-2 cells. RT-qPCR analysis (**j**) and WB analysis (**k**) of expression of indicated genes and proteins in PC-3 cells stably expressing the indicated shRNAs. Meta-analysis of RNA-seq data showing the association of increased expression of *RB1* (**l**) or *GNB1L* (**m**) with the WT or deletion statuses of *RB1* gene in PCa samples of the TCGA cohort. **n** The correlation between *GNB1L* and *RB1* mRNA expression levels in PCa samples of the TCGA cohort. **o** Kaplan–Meier Survival curve showing the association of high *GNB1L* mRNA expression with poor overall survival of PCa samples of the TCGA cohort. Western blot assays in **i**, **k** were repeated two independent times with similar results. Data in **e**–**h** and **j** were shown as mean ± s.d. from three independent experiments with two-sided Student's *t* test for the statistical analysis. Data was performed by two-sided Fisher's exact test from Enrichr (https://maayanlab.cloud/Enrichr/) in (**c**). Statistical analysis for data in **a**, **l**, **m**, **n**, and **o** was indicated in the method section.

These data support the observation reported previously[30,31] that loss of RB promotes antiandrogen resistance and neuroendocrine phenotype. However, none of these effects was mediated through RB loss-induced upregulation of GNB1L. Of course, our data cannot completely rule out the possibility that other GPCR-cAMP-related genes upregulated by RB loss-induced aberrant activation of BRD4 may contribute to antiandrogen resistance and neuroendocrine phenotype in RB-deficient cells. E2F1 knockdown did not alter GNB1L expression in both RB-proficient and -deficient PC-3 cells (Supplementary Fig. 8a, b), indicating that its expression is not regulated by the RB-E2F1 axis. Together, our data show that RB deficiency induces upregulation of GNB1L and activation of the GPCR-cAMP-CREB axis in PCa cells and that this effect is mediated through BRD4.

## RB S249/T252 phosphorylation promotes GNB1L expression and CREB activation

Since RB S249/T252 phosphorylation inhibits RB-BRD4 interaction, we sought to determine whether S249/T252 phosphorylation regulates GNB1L expression via BRD4. Consistent with the regulation of GNB1L expression by BRD4 knockdown, we found that in a dose-dependent manner treatment of PC-3 cells with the BRD4 inhibitor JQ1 inhibited GNB1L expression at both mRNA and protein levels (Supplementary Fig. 8c, d). As RB-BRD4 interaction can be enhanced by RB S249/T252 dephosphorylation, we examined whether the expression of GNB1L and activation of CREB were influenced by RB dephosphorylation. We demonstrated that activation (dephosphorylation) of RB by treatment with CDK4/6 inhibitor palbociclib decreased GNB1L expression at protein and mRNA levels and inhibited CREB phosphorylation in a dose-dependent fashion in both PC-3 and C4-2 cell lines (Fig. 5a, b and Supplementary Fig. 8e, f). In agreement with this observation, ChIP-qPCR assay showed that BRD4 binding on chromatin at the *GNB1L* gene locus was attenuated after palbociclib treatment in both cell l lines (Fig. 5c and Supplementary Fig. 8g). Furthermore, we showed that ectopic expression of WT RB suppressed GNB1L mRNA and protein expression in DU145 cells in which endogenous RB is malfunctional (Fig. 5d, e). Compared to the WT RB, the inhibitory effect of RB S249/T252 phosphorylation-resistant mutant RBΔCDK was much greater as evident by the reduced expression of GNB1L protein and CREB phosphorylation of CREB (Fig. 5d, e).

To verify these findings, we knocked down endogenous RB and performed rescue experiments. We found that RB knockdown-induced upregulation of GNB1L expression and CREB phosphorylation was reversed by restored expression of shRNA-resistant WT RB and this effect was largely enhanced by expression of phosphorylation-resistant mutant RBΔCDK (Fig. 5f, g). In line with these observations, enhanced expression of CDK4 and Cyclin D1 increased GNB1L expression and induced BET inhibitor resistance in C4-2 and LNCaP cells (Fig. 5h–k). In various PCa cell lines with different levels of total and phosphorylated RB, we further demonstrated that PC-3 and DU145 cell lines (in which little/no RB protein or high level of phosphorylated RB was expressed) were much insensitive to BET inhibitors compared to C4-2 and LNCaP (in which low level of phosphorylated RB was expressed) (Supplementary Fig. 8h–j). Notably, these results are consistent with the previous report that DU145 and PC-3 cells were relatively insensitive to JQ1 compared to LNCaP cells[32]. We also performed immunochemistry (IHC) analysis using a tissue microarray (TMA) containing 115 cores resulting from 51 patients with advanced PCa to investigate the correlation between RB phosphorylation and GNB1L expression. We found that high expression of GNB1L positively correlated with elevated RB S249/T252 phosphorylation and their correlation was statistically significant (Fig. 5l, m). Collectively, our data suggest that RB S249/T252 phosphorylation can promote GNB1L expression in PCa cells in culture and patient samples.

## CREB inhibitor 666-15 overcomes JQ1 resistance in RB-deficient PCa cells in vitro and in vivo

Next, we sought to determine whether targeting GPCR-cAMP signaling enables to overcome BET inhibitor resistance in RB-deficient PCa cells. To this end, we treated control or RB KD PC-3 cells with JQ1 or CREB inhibitor 666-15 individually or both prior to MTS assays. We demonstrated that while RB KD cells were highly resistant to JQ1, both control and RB KD cells were equivalently sensitive to 666-15 (Fig. 6a). Importantly, 666-15 co-treatment completely abolished JQ1 resistance in RB-deficient cells (Fig. 6a). Similar results were obtained in C4-2 cells (Supplementary Fig. 8k). Using colony formation assay we further validated that co-treatment of 666-15 completely abolished JQ1 resistance in these two cell lines (Fig. 6b–d and Supplementary Fig. 8l, m).

We also examined the anti-cancer effect of JQ1 and the CREB inhibitor in RB-deficient tumors in vivo. As expected, RB depletion largely enhanced PC-3 tumor growth in mice (Fig. 6e–g). Importantly, RB-deficient PC-3 tumors were resistant to JQ1 in comparison to control tumors (Fig. 6e–g), consistent with the results obtained in PC-3 cells cultured in vitro (Fig. 6a). However, co-administration of 666-15 abolished JQ1 resistance of RB-deficient PC-3 tumors (Fig. 6e–g), suggesting that there was seemingly no obvious drug-drug interaction between these two compounds under these conditions. Analysis of Ki67 and cleaved Caspase-3 IHC in tumor tissues showed that JQ1 treatment only resulted in a minimal effect on proliferation of RB-deficient tumors, but Ki67 expression was largely inhibited by the combined treatment of JQ1 and 666-15 (Fig. 6h, i). Although JQ1 treatment alone had no obvious effect on the expression of cleaved Caspase-3, an indication of apoptosis, the combination of JQ1 and 666-15 drastically induced expression of cleaved Caspase-3 (Fig. 6j, k). In agreement with the result of cell death, there was almost no effect of JQ1 treatment on expression of BCL-2, a known transcription target of CREB and a well-studied anti-apoptotic protein[33] (Fig. 6d). In contrast, treatment with 666-15 in combination with JQ1 completely abolished RB depletion-induced upregulation of BCL-2 (Fig. 6d). Together, these data indicate that activation of CREB plays an essential role in mediating BET inhibitor resistance in RB-deficient PCa cells and that inhibition of CREB overcomes BET inhibitor resistance.

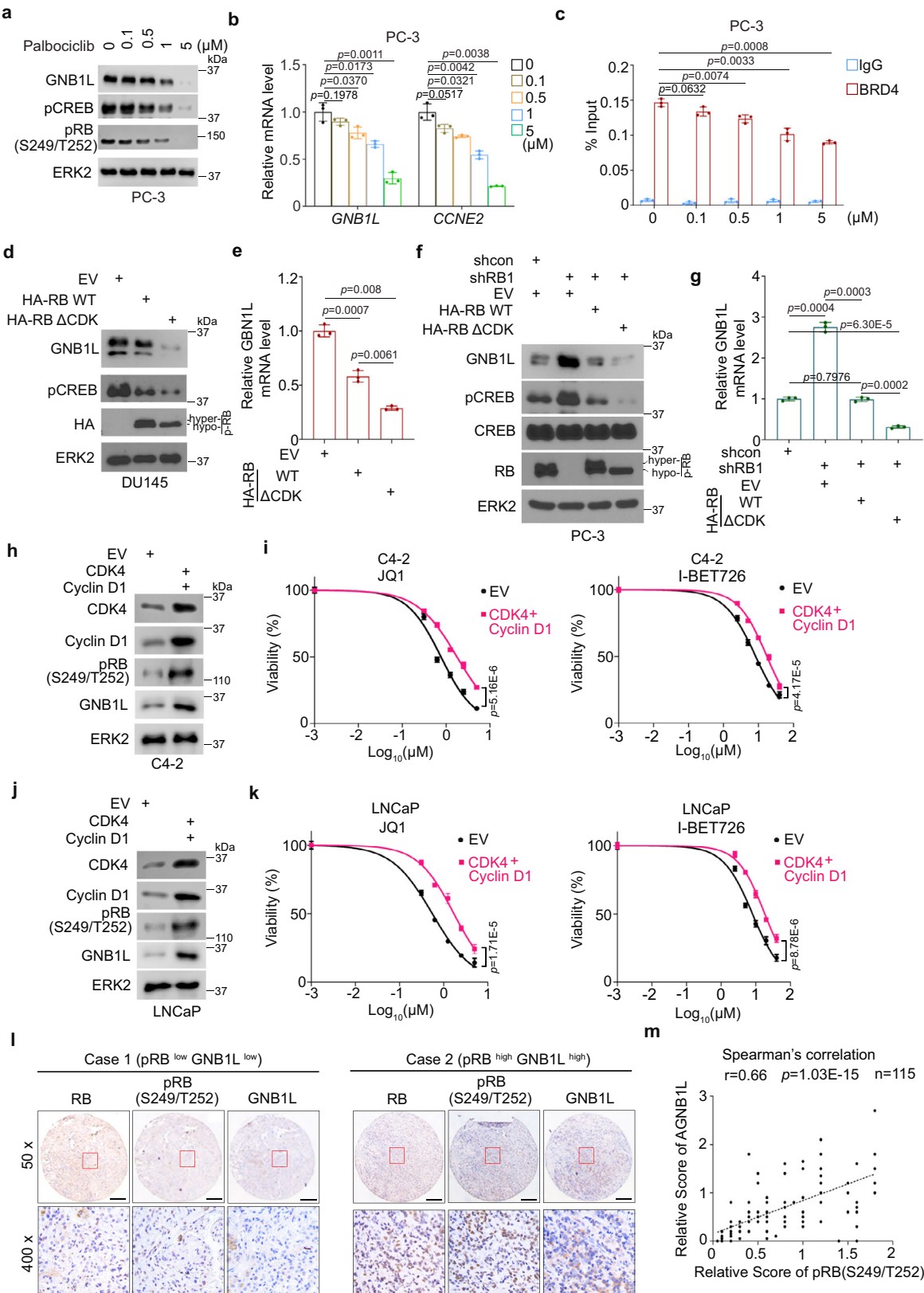

## Discussion

Through a small-scale screening of signaling pathway inhibitors such as BET inhibitor, CBP/p300 inhibitor, PARP inhibitor, and AKT inhibitor, we uncovered that RB-deficient PCa cells are resistant to BET inhibitors. It has been shown previously that RB-null NUT midline carcinoma cells are resistant to BET inhibitors and the resistance is

believed to be mediated through the RB-E2F1 pathway[23]. Different from the finding in NUT carcinoma cells, however, we provide evidence that E2F1 knockdown had little or no obvious effect on the BET inhibitor sensitivity in both RB WT and deficient PCa cells. This difference can be explained, at least in part by the differential effects of JQ1 on the expression of E2F1 in these two cancer types. While JQ1

**Fig. 5 | RB dephosphorylation downregulates GNB1L expression and inhibits CREB pathway.** WB (**a**) and RT-qPCR (**b**) analysis of GNB1L protein expression in PC-3 cells treated with vehicle or different doses of palbociclib for 36 h. **c** ChIP-qPCR analysis of BRD4 occupancy in the *GNB1L* gene locus in PC-3 cells treated with vehicle or different doses of palbociclib. WB (**d**) and RT-qPCR (**e**) analysis of GNB1L protein expression in DU145 cells transfected with HA-RB or HA-RB (ΔCDK). WB (**f**) and RT-qPCR (**g**) analysis of expression of GNB1L protein in indicated PC-3 cells. WB analysis in C4-2 (**h**) and LNCaP (**j**) cells transfected with empty vector (EV) or Cyclin D1 and CDK4 for 24 h followed by cell viability analysis (**i**, **k**) after treatment with different doses of BET inhibitors for 72 h. Representative images of GNB1L and phosphorylated RB at S249/T252 (pRB) IHC staining on a TMA from a cohort of metastatic PCa patient samples. Scale bar in '50 x' represented 150 μm while that in '400 x' represented 50 μm. **m** Positive correlation between GNB1L and pRB protein expression in the cohort of PCa patient samples. Data in **b**, **c**, **e**, and **g** were as mean ± s.d. from three independent experiments. The statistic test was performed by two-sided Student's *t* test. Data in **i** and **k** were from five replicates. Data in **i** and **k** were analyzed by two-way ANOVA. The Spearman's correlation was performed in (**m**).

treatment decreased E2F1 expression in NUT carcinoma cells, no such effect was observed in PCa cells, reinforcing the notion that BET inhibitor resistance in PCa cells is mediated through the mechanism independent of E2F1. It has been shown previously that the CDK4/6 inhibitor palbociclib can largely sensitize cells to JQ1 treatment in RB-proficient cells[23,34,35]. However, such combination is not applicable in RB-deficient PCa cells, highlighting that a new strategy is needed to overcome BET inhibitor resistance. By identifying RB protein as a binding partner and an intrinsic inhibitor of BRD4, we further show that loss of RB results in aberrant occupancy of BRD4 in the loci of GPCR-cAMP pathway genes. Importantly, we convincingly show that co-targeting this signaling pathway overcomes BET inhibitor resistance in RB-deficient cells. Hence, our work not only provides molecular insights into the mechanism of BET inhibitor resistance in RB-deficient PCa cells, but also define a viable therapeutic strategy to overcome the resistance.

We and others have previously reported a FXXXV-containing motif in client proteins mediating the binding by RB-N[19,20]. While there is a similar FXXXV motif in all non-testicular BET proteins including BRD2, BRD3, and BRD4, to our surprise, such motif is not required for BET protein binding of RB-N. Instead, we identify a FXXXI motif ([157]FLQKI[161]) as an RB-recognized motif in the BD1 domain of BRD4. Additionally, quite a few glutamic acid residues are present surrounding the [157]FLQKI[161] motif. Since these residues are negatively charged and it is not surprising that this region tends to bind to the positively charged arginine (R)-rich linker of RB-N. Due to the same reason, phosphorylation of S249/T252 in the R-linker region in RB-N, which induces negative charges, impairs RB-N interaction with BD1 of BRD4. Furthermore, the residues surrounding the [87]FQQPV[91] motif in BD1 of BRD4 are less acidic than that in the [157]FLQKI[161] region, which may provide a plausible explanation as to why the binding of this motif to RB-N is relatively weaker despite it is a typical FXXXV motif. Taken together, our study identifies a previously unrecognized FXXXI motif in mediating the client protein binding of the R-linker region in RB-N (Fig. 7). Our findings reinforce the notion that amino acid composition and the charges surrounding the FXXXI or FXXXV motif in the client proteins are the key determinants for their tight or loose interaction with RB-N. Reciprocally, this model is also consistent with our finding that phosphorylation of S249/T252 in the R-linker region of RB-N by CDK4/6 weakens the interaction between BRD4 BD1 and RB-N (Fig. 7).

Both BD1 and BD2 are shown to be important for BRD4 binding of acetylated histones and/or chromatin-associated factors[3,27]. Our ChIP-seq data analysis show that RB binding primarily limits the chromatin binding capacity of BRD4 (Fig. 7) since BRD4 binding on chromatin is largely increased, but only decreased in a handful of loci in the genome upon RB knockdown. Among the genes with gained occupancy of BRD4, GPCR-cAMP signaling pathway genes such as *GNB1L* are upregulated upon RB depletion. GNB1L belongs to the family of guanine nucleotide-binding proteins and has been shown to activate the cAMP-PKA-CREB signaling cascade[36]. Importantly, we found that GNB1L-activated CREB confers resistance to BET inhibitors in RB-deficient PCa cells. CREB inhibitor has been suggested as a promising anti-cancer agent in multiple cancer types including acute myeloid leukemia[37], pancreatic cancer[38], and breast cancer[39]. Our findings in cultured PCa

cells and xenograft tumors in mice invariably suggest that CREB is also a potential therapeutic target of PCa, especially those in which the *RB1* gene is deleted or protein becomes hyperphosphorylation due to aberrant activation of CDK4/6 (Fig. 7).

In summary, our work identifies RB protein as a *bono fide* endogenous inhibitor of BRD4 and demonstrates that RB loss or phosphorylation at S249/T252 by CDK4/6 confers PCa cell resistance to small molecule BET inhibitors by upregulating GNB1L expression and activating GNB1L-CREB signaling cascade. Our results imply that loss of RB, aberrant activation of CDK4/6 as well as CREB high expression could serve as biomarkers to predict the BET inhibitors in PCa. Furthermore, our studies suggest that RB-deficient cells become susceptible to BET inhibitors after co-administration of the CREB inhibitor. Therefore, our findings not only illustrate the molecular mechanism of BET inhibitor resistance caused by RB loss, but also reveal a potential therapeutic strategy for prostate cancers with abnormalities in RB, CDK4/6, and/or CREB signaling.

## Methods
### Antibodies, plasmids, and chemicals
The following antibodies are used in the experiments: Anti-RB (1:2000 in dilution, 554136, BD bioscience), p107/RBL1 (1:1000 in dilution, SC-318, Santa Cruz Biotechnology), p130/RBL2 (1:1000 in dilution, SC-317, Santa Cruz Biotechnology), ERK2 (1:2000 in dilution, SC-1647, Santa Cruz Biotechnology), HA (1:1000 in dilution, 901515, Biolegend), FLAG (1:1000 in dilution, F-3165, Sigma), V5 (1:1000 in dilution, SC-81594, Santa Cruz Biotechnology), CDK4 (1:1000 in dilution, SC-601, Santa Cruz Biotechnology), CDK6 (1:1000 in dilution, SC-177, Santa Cruz Biotechnology), Histone H3 (1:1000 in dilution, 9715, Cell Signaling Technology), BRD2 (1:1000 in dilution, ab139690, Abcam), BRD3 (1:1000 in dilution, A302-368A, Bethyl), BRD4 (1:1000 in dilution, A301-985A100, Bethyl), BRD4 (1:1000 in dilution, ab128874, Abcam). GNB1L (1:1000 in dilution, HPA034627, Sigma), Cleaved PARP (1:1000 in dilution, 5625, Cell Signaling Technology), Cleaved Caspase-3 (Asp175) (1:1000 in dilution, 9661, Cell Signaling Technology), p50 (1:1000 in dilution, 13586, Cell Signaling Technology). Secondary antibodies including anti-mouse (light chain) secondary antibody were purchased from Jackson Immunoresearch at 1:5000 of dilution when used. Plasmids for HA-RB and related mutants, HA-Cyclin D1 and V5-CDK4 are described previously[19,40]. FLAG-GNB1L plasmid was purchased from GenScript (#OHU05623D). Detailed information of the shRNAs and sgRNAs used in this study is shown in Supplementary Data 4. LentiCRISPR v2-dCas9 plasmid was obtained from Addgene (#112233). JQ1 was purchased from TargetMol (#T2110). I-BET726 was obtained from Cayman (#16872). 666-15 was purchased from Millipore (#538341). Palbociclib was purchased from APExBIO (#A8316).

### Cell lines and cell culture
PC-3, DU-145, LNCaP, and 293T cells were purchased from American Type Culture Collection (ATCC). C4-2 cells were purchased from Uro Corporation. PC-3, DU145, LNCaP, and C4-2 cells were cultivated in

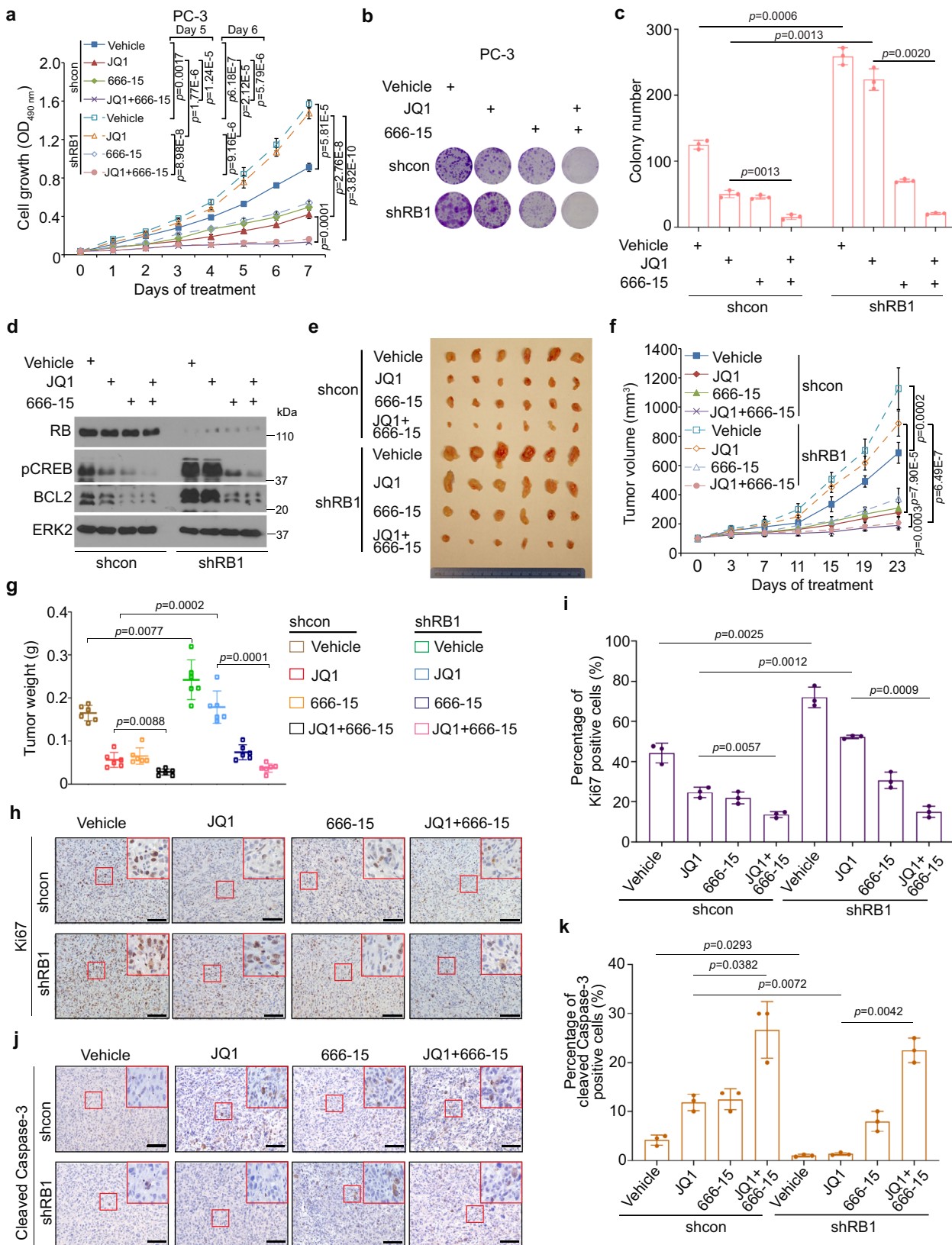

RPMI 1640 media (Corning) with 10% fetal bovine serum (FBS) (Invitrogen/Thermo Fisher Scientific). 293T cells were grown in DMEM media (Corning) supplemented with 10% FBS (Millipore). All the cells were incubated at 37 °C supplied with 5% $CO_2$. Cells were treated with plasmocin (Invivogen) to anti-mycoplasma before the subsequent experiments.

## Transfection and lentivirus infection

Lipofectamine 2000 (Thermo Fisher Scientific) was used for transient transfection with indicated plasmids according to the manufacturer's instruction. The supernatant containing lentivirus was collected from 293T cells co-transfected with plasmids of psPAX2, pMDG2 and control or gene-specific shRNAs and filtered using 0.22 μm filter

**Fig. 6 | CREB inhibitor treatment overcomes BET inhibitor resistance in RB-deficient PCa cells in vitro and in vivo. a** Growth of control and RB knockdown PC-3 cells treated with JQ1 (1 μM) in combination with or without CREB inhibitor 666-15 (1 μM) for different periods of time. Statistical analyses were performed for data at Day 5, 6, and 7 time points. Representative image (**b**) and quantification of the data (**c**) of cell colony formation assay performed in control and RB knockdown PC-3 cells treated with indicated inhibitors for 14 days. **d** WB analysis of indicated proteins in control and RB knockdown PC-3 cells. **e** Representative images of tumors isolated from mice at 23 days after the indicated treatment. **f** Tumor growth curve in mice treated with vehicle or indicated inhibitors. **g** Weight of tumors from mice at 23 days after treatment with vehicle or indicated inhibitors. **h–k** IHC images and quantitative data of Ki67 (**h**, **i**) and cleaved Caspase-3 (**j**, **k**) in xenografts in mice as treated in (**e**). Scale bar in the IHC images in **h**, **j** represented 100 μm. Data in **a** shown as means ± S.D. from five replicates. Data in **c** shown as means ± S.D. from three independent experiments. Data in **f** and **g** shown as means ± S.D. from six tumors (*n* = 6) in each group. Data in **i** and **k** shown as mean ± s.d. from three independent experiments. For each experiment, five independent fields were enrolled for the calculation. Two-sided Student's *t* test was used for **a**, **c**, **g**, **i**, and **k**. Two-way ANOVA was performed in (**f**).

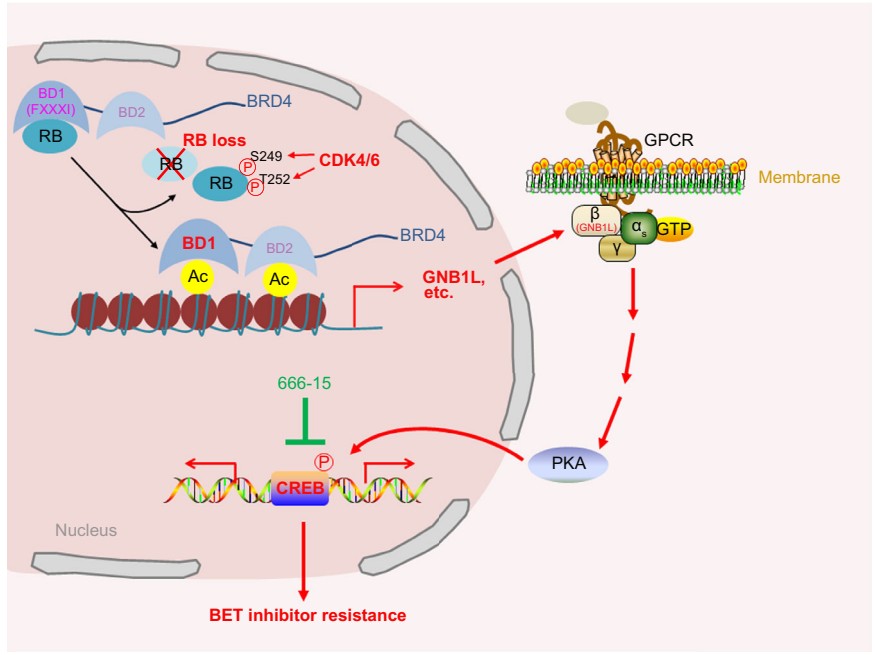

**Fig. 7 | The schema depicting the BET inhibitor resistance in PCa cells due to RB loss and hyperphosphorylation.** RB binds to the BD1 of BRD4 and impedes BRD4 targeting of chromatin. However, RB loss or phosphorylation by CDKs such as CDK4/6 augments BRD4 occupancy on chromatin at genomic loci such as genes in the GPCR-cAMP signaling pathway, thereby leading to aberrant activation of CREB and BET inhibitor resistance. The resistance can be overcome by co-targeting of CREB, representing a vulnerability for BET inhibitors in RB-deficient cancer cells.

(Millipore). The filtered virus was added to the indicated cells and infected cells were selected with puromycin (1.5 μg/mL).

### Cell proliferation assay
For IC50 assay, DU145 (1000/well), PC-3 (1000/well), C4-2 (2000/well) or LNCaP cells (2000/well) were seeded in 96-well plates overnight. Cells were then treated with different doses of the indicated compounds. At 72 h after treatment, MTS reagent (Promega) was added to each well and the plates were incubated at 37 °C for 2 h. Optical density (OD) of cells was measured at 490 nm using microtiter reader (Biotek). For cell growth curve assay, PC-3 cells (800/well) or C4-2 cells (1500/well) were seeded in 96-well plates overnight and treated with indicated chemicals. At the indicated time points, MTS reagent (Promega) was added to each well and the plates were incubated at 37 °C for 2 h. Optical density (OD) of cells was measured at 490 nm using microtiter reader (Biotek).

### RNA extraction and reverse transcription-quantitative polymerase chain reaction (RT-qPCR)
The total RNA was isolated from cultured cells using Trizol reagent (Thermo Fisher Scientific) according to the manufacturer's instructions. Complementary DNA was synthesized using reverse transcriptase kit (Promega). Two-step real-time PCR was performed using the SYBR Green Mix (Bio-Rad) and CFX96 Real-Time System on C1000 Touch Thermal Cycler (Bio-Rad) according to the manufacturer's instructions. Relative gene expression was normalized to the expression of house-keeping gene glyceraldehyde-3-phosphate dehydrogenase (*GAPDH*). The sequence information of the primers used for RT-qPCR are provided in Supplementary Data 4.

### Co-immunoprecipitation (co-IP) assay
Cells were collected after transfected with indicated plasmids. After washing with 1 × PBS, cells were lysed in IP buffer (0.5% NP-40, 20 mM Tris-HCl, pH = 8.0, 10 mM NaCl, 1 mM EDTA) supplied with a cocktail of protease inhibitors (Sigma-Aldrich). Cell lysate was immunoprecipitated with indicated antibodies against target proteins in the presence of protein A/G beads (Millipore) overnight. Beads were washed with IP buffer for five times and boiled in protein loading buffer (Bio-Rad) for further analysis by Western blot.

### Chromatin immunoprecipitation (ChIP), high throughput sequencing and ChIP-qPCR
Control and RB-knockdown PC-3 and C4-2 cells were treated with disuccinimidyl glutarate (DSG) (A35392, Thermo Fisher Scientific) for 30 min at room temperature before being fixed in 1% formaldehyde. Cells were quenched with 2.5 μM glycine and collected for sonication. The supernatant was obtained after centrifuge and mixed with protein A/G beads and BRD4 antibodies (10 μg). After incubation overnight, beads were washed. DNA-protein complex was eluted and reverse crosslink was performed at 65 °C for 2 h. The elution was further

treated with RNase A and proteinase K. Enriched DNA was collected for high throughput sequencing (ChIP-seq) or quantitative PCR (ChIP-qPCR). For ChIP-seq, the high-throughput sequencing was performed using the Illumina HiSeq 4000 platform by Genome Analysis Core at Mayo Clinic. Primers for different BRD4 target genes were designed based on the BRD4 binding peaks identified by ChIP-seq. BRD4 binding of specific target loci were validated by ChIP-qPCR using the SYBR Green Mix (Bio-Rad) and CFX96 Real-Time System on C1000 Touch Thermal Cycler (Bio-Rad) according to manufacturer's instructions. The sequence information of the primers used for ChIP-qPCR are provided in Supplementary Data 4.

## ChIP-seq data analysis
The raw reads of ChIP-seq were mapped to the human reference genome (GRCh37/hg38) using bowtie2 (version 2.2.9). MACS2 (version 2.1.1) was run to perform the peak calling with a $p$ value threshold of $1 \times 10^{-5}$. BigWig files were generated for visualization using the UCSC Genome Browser. The assignment of peaks to potential target genes was performed by the Genomic Regions Enrichment of Annotations Tool (GREAT). DiffBind software from an open source (https://bioconductor.org/packages/release/bioc/html/DiffBind.html) was utilized to determine the differential binding of BRD4 in the genome of the control and RB knockdown C4-2 cells. Data was collected for differential analysis by DEseq2, calculated for $p$ value by two-sided Wald test after modeling the count data by logistic regression, and $p$ value was adjusted by the Benjamini and Hochberg method for multiple comparison. The ChIP-seq data have been deposited into the NCBI GEO data repository with the accession code GSE191263.

Cistrome database BRD4 ChIP-seq samples, the peak files and QC metrics were downloaded by batch download function at http://cistrome.org/db/#/bdown. In total, 294 human BRD4 ChIP-seq samples available in the batch download regardless of cell type or disease type. CHIPs pipeline was used to run BRD4 ChIP-seq samples, and QC report was generated by CHIPs pipeline (https://liulab-dfci.github.io/resources/publications/F1000Rsch10_517.pdf).

## Preparation of chromatin fraction
PC-3 cells transfected with the indicated plasmids were collected for the fractionation as described[41]. After washing with cold $1 \times$ PBS, cells were incubated with lysis buffer (50 mM Tris-HCl, pH 7.5, 5% glycerol, 150 mM NaCl, 1.5 mM MgCl$_2$) supplied with 0.5 % NP-40 at 4 °C for 30 min. The lysate was centrifuged at $1500 \times g$ for 5 min. The supernatant was then spun at 20,000 g for 15 min and the soluble fraction was collected. The pellet from the first-time centrifugation was resuspended in 2 volumes of lysis buffer supplemented with 0.2% SDS and 1:1000 dilution of Benzonase Nuclease (E1014, Sigma-Aldrich) after washing one time with lysis buffer, and incubated in an incubator shaker at 37 °C at 850 rpm for 30 min. The chromatin fraction was spun down by 20,000 g for 15 min. The soluble and chromatin fractions were further analyzed by Western blot.

## Crystal violet staining
PC-3 (1,000) or C4-2 cells (2000) were seeded in 6-well plates overnight. PC-3 cells were treated with 1 μM JQ1 and 1 μM 666-15 individually or both. C4-2 cells were treated with 0.5 μM JQ1 and 0.5 μM 666-15 individually or both. After 14 days of treatment, cells were washed with $1 \times$ PBS and fixed with 4% paraformaldehyde (Sigma-Aldrich) for 20 min at room temperature. After washing with $1 \times$ PBS for twice, cells were stained with crystal violet (0.5 g crystal violet in 80 ml H$_2$O with 20 ml methanol) for 30 min at room temperature. Cells were washed with distilled water gently to remove the remaining crystal violet.

## GST-tagged recombinant protein purification
Plasmids for GST-tagged recombinant proteins were transformed into *E. coli* BL21. The successfully transformed BL21 cells were cultured in

flasks in an incubator shaker and treated with 100 μM IPTG (Sigma-Aldrich) at 18 °C overnight. After IPTG induction, bacteria were collected and resuspended in lysis buffer (50 mM Tris-HCl, pH 8.0) suppled with protease inhibitor (Sigma-Aldrich) and sonicated. Glutathione Agarose (Thermo Fisher Scientific) was added to enrich the GST-tagged protein. The 10 mM reduced glutathione (Sigma-Aldrich) in 50 mM Tris-HCl, pH 8.0 was added and incubated with agarose for 1 h at room temperature. The eluted protein was collected by centrifuge and saved at −80 °C for further use.

## In vitro kinase assay
Cyclin D1 and CDK4 proteins were synthesized using the in vitro transcription and translation system (L1170, Promega) according to the manufacturer's instructions. These proteins were then incubated with the mixture of GST-RBN or GST-RBN (S249A/T252A) proteins and GST beads in the kinase reaction buffer (25 mM HEPES pH 7.5, 25 mM β-glycerophosphate, 25 mM MgCl$_2$, 2 mM dithiothreitol, 0.2 mM ATP and 0.1 mM NaVO3) at 30 °C for 2 h. After washing with wash buffer (25 mM HEPES pH 7.5, 150 mM NaCl, 1.5 mM MgCl$_2$) twice, the beads were incubated with lysate of 293T cells at 30 °C for 2 h. The beads were washed with wash buffer for five times, and beads were boiled in sample loading buffer and subjected to the Western blot analysis.

## Prostate cancer patient samples
With the approval of the institutional review board (IRB), metastatic PCa specimens were obtained from patients undergoing standard-of-care biopsies at Mayo Clinic (Rochester, MN). Informed consent was obtained for use of the samples for research by the time the specimens were collected by the Mayo Clinic. A tissue microarray was constructed from the formalin-fixed paraffin-embedded (FFPE) samples of metastatic PCa identified after a search of pathologic and clinical databases of archival tissues. The Mayo Clinic institutional review board approved the experimental protocols for retrieving pathology blocks/slides and accessing electronic medical records. Cores in which greater than 50% of the tissue was lost during IHC were excluded from analysis. A total of 115 tissue microarray cores resulting from 53 samples (20 bone metastases and 33 nonbone metastases) of 51 patients were used for analysis. The study design and conduct complied with all relevant regulations regarding the use the specimens of human study participants. The study was conducted in accordance to the criteria set by the Declaration of Helsinki.

## Generation and treatment of prostate cancer xenografts in mice
Mouse experiments were approved by the Institutional Animal Care and Use Committee (IACUC) at the Mayo Clinic. Six-week-old male SCID mice were generated in house. Control and RB knockdown PC-3 cells ($5 \times 10^6$) mixed with Matrigel mixture ($1 \times$ PBS: Matrigel (BD Biosciences) = 1:1) were injected subcutaneously into SCID mice. After the average size of the xenografts reached approximately 100 mm$^3$, mice were treated intraperitoneally with vehicle (10% β-cyclodextrin (Sigma-Aldrich), 1% N-methylpyrrolidone (NMP) (Thermo Fisher Scientific), 5% Tween-80 (Sigma-Aldrich) in $1 \times$ PBS), JQ1 (Sigma-Aldrich, dissolved in 10% β-cyclodextrin) at 50 mg/kg or 666-15 (TargetMol, dissolved in 1% N-methylpyrrolidone (NMP), 5% Tween-80 in $1 \times$ PBS) at 10 mg/kg 5 days a week. Tumor length ($L$) and width ($W$) were measured every 4 days, and tumor volumes were calculated by the formula $(L \times W^2)/2$. Mice were sacrificed by euthanasia and tumors were collected and photographed. One portion of the tumor tissues was used for FFPE and the rest was frozen for RNA and protein extraction.

## Immunohistochemistry (IHC)
The FFPE xenograft tissues and patient specimens were consecutively cut at 4 μm. The tissues on slides were rehydrated, the activity of endogenous peroxidase was inhibited and antigen retrieval was performed as previously described[9]. The tissues were incubated with

primary antibody overnight at 4 °C. The following primary antibodies were used: anti-Ki67 (ab15580, Abcam) and anti-cleaved Caspase-3 (9661, Cell Signaling Technology). After washing, the tissues were incubated with secondary antibody (BA1000, Vector Lab) for 1 h at room temperature. After counterstaining, the tissues were dehydrated and covered with a coverslip. For quantification, cells with positive staining in the nucleus were identified and included to calculate the percentage of Ki67 positive-staining cells.

### Meta-analysis of patient data
The patient data from TCGA database was obtained from cBioPortal (https://www.cbioportal.org/). The Z-score (FPKM) for mRNA level of each gene of interest was downloaded from the database for further analysis. $N = 267$ for the RB 'WT' group and $n = 218$ for the RB 'Deletion' group. Data were shown as mean ± s.d., and two-tailed Mann–Whitney U test was carried out to calculate $p$ value for the comparison. Relative level of GNB1L and RB expression in 485 patients was included for correlation analysis, which was determined by two-tailed Spearman's rank correlation coefficient. Log-rank (Mantel–Cox) test was performed to determine the statistical differences between stratified groups used for Kaplan–Meier Survival curve analyses.

### Statistical analysis
$P$ values were determined by a two-tailed Student's t test, two-way ANOVA test, log-rank test, Mann–Whitney U test. All data are shown as mean values ± S.D. for experiments representing three independent experiments except particular indication. $P$ values < 0.05 were considered statistically significant.

### Reporting summary
Further information on research design is available in the Nature Research Reporting Summary linked to this article.

## Data availability
The data that support this study are available from the corresponding authors upon reasonable request. The ChIP-seq data generated in this study have been deposited into at the NCBI's GEO data repository with the accession code GSE191263. The Human reference genome (GRCh38/hg38) (GenBank/RefSeq assembly accession numbers GCA_000001405.15/GCA_000001405.26) was used for mapping of ChIP-seq reads. Source data are provided with this paper.

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

## Acknowledgements

This work was supported in part by grants from the National Institutes of Health (CA134514, CA130908, CA193239, and CA203849 to H.H.) and the Mayo Clinic Foundation (to H.H.).

## Author contributions

H.H. conceived the study. D.D and L.S. performed experiments, data collection, and analysis. R.J. supervised histological and IHC analysis in patient samples. X.H. and S.J.W. provided reagents and participated in experimental design. L.W., R.Z., and Y.T. performed bioinformatics analysis. H.H., L.S., and D.D. wrote the manuscript.

## Competing interests

The authors declare no competing interests.
