## [Peer Review File · Nature Communications]

REVIEWER COMMENTS

Reviewer #1 (Remarks to the Author):

In this manuscript the authors show that RB1 mediates response to BET bromodomain inhibitors due to its direct interaction with BRD4. The authors mapped this interaction to the N-terminal part of RB1 and showed that RB1 phosphorylation by CDK4/6 kinase disrupts this binding. Based on ChIPseq for RB1 and BRD4 the authors determined that RB1 deletion enhances BRD4 chromatin binding to certain loci, and many of these encode genes involved in cAMP signaling. Subsequently they showed that inhibition of cAMP/CREB signaling restored BET inhibitor responsiveness of RB1 deficient cancer cells.

Specific points:

1. The experiments are done with a single cell line, PC3 cells. Although there are limited prostate cancer cell lines available, it would be important to reproduce the key findings in other lines.
2. Most protein-interaction experiments are done with exogenously overexpressed tagged proteins. Validation by using endogenous proteins is important.
3. The methods section is very limited, many experiments are missing and even the ones listed are not described in sufficient detail to evaluate them.
4. The statistical tests are missing for all figures, the test used to calculate p-values have to be indicated for each figure.
5. The quality of the BRD4 ChIPseq data seems questionable based on Figure 4. The authors should run QC in CISTROME to ensure the reliability of the data. Since many downstream experiments depend on this ChIPseq data, it is a key data in the manuscript.
6. The authors should provide an irrelevant chromatin-bound protein as control for the co-IP experiments, since looks like everything that's on the chromatin is detected in the RB co-IP in Figure 2.
7. cAMP signaling is commonly associated with neural features and acquisition of neuroendocrine phenotype is a mechanisms of anti-androgen resistance in prostate cancer. Do the tumors with low RB and high GNB1L expression more likely to be resistant to anti-androgens and have neuroendocrine features? Based on the association with survival (Fig 4o) this is a possibility.
8. The concentration of Palbociclib used (Figure 5a) is very high and the authors only see an affect at 5uM. This is not usual. Also the legend of Fig5a is shifted and the text is not above the lanes.
9. The cell proliferation growth assay is too short (Fig 6a), the authors should extend this until at least 3 independent points show statistically significant differences. right now only day 5 is different, so they need to go up to day 10-12. It's also unclear how they have less than 1 day of treatment – probably the x-axis label is messed up?
10. The graphs showing the cell proliferation and tumor volume differences are very busy and hard to differentiate the colors.
11. Please do NOT use bar graphs, show the data as dot plots to each data point can be

seen. This applies to Fig 6c, 5g, 6g, and many others.

Reviewer #2 (Remarks to the Author):

In this study, Ding et al explore the relationship between the RB and BRD4, specifically as it relates to resistance to BET inhibitor (BETi) drug treatment. The authors first created RB +/- prostate cancer cell lines which they tested against numerous clinically utilized drugs, finding that RB loss promoted resistance to BETi in an E2F1 independent manner. The authors further explored the relationship between RB and BRD4 through a number of binding studies, ultimately elucidating the interaction between the two proteins and the effect of RB phosphorylation on this interaction. Through BRD4 ChIP-Seq studies in the RB k/d lines, Ding et al nominated a number of putative target genes within the cAMP pathway for downstream characterization, ultimately showing GNB1L to be sensitive to changes in RB and BRD4 status. Finally, the authors show that the resistance to BETi seen in the RB k/d models can be overcome with combination treatment of the CREB inhibitor 666-15. This paper is well written and has a logical design. Additionally, the findings presented by the authors are noteworthy and have the potential to significantly influence prostate cancer clinical practice as well as future clinical trial opportunities in this space.

Comments:

1. The authors primarily utilize PC-3 and C42 cell lines for the RB +/- experiments. As recent studies have suggested HTS and CRPC cell lines have differential effects of RB loss (Mandigo et al 2021) it would be beneficial to see whether the same effects are seen in a HTS line such as LNCaP.
2. The experiments which evaluated the role for E2F1 in RB loss-induced resistance to JQ1 utilized AR-negative models. Can these experiments be recapitulated in the AR-positive C42 lines used earlier in the paper?
3. What method was utilized to determine the differential binding for the ChIP-Seq studies?
4. The authors show that GNB1L expression patterns are similar in TCGA data, was this also true for the other genes in the cAMP pathway that were identified through the ChIP-Seq studies?
5. While the authors utilized RB/BRD4 alteration to toggle GNB1L expression, it would be interesting to how GNB1L overexpression alone affects the phenotypic results including resistance to BETi
6. Can the authors comment on the potential drug-drug interaction between JQ1 and the CREBi compound used in the study?

Reviewer #3 (Remarks to the Author):

This study identifies a novel mechanism by which BET inhibitors develop resistance in prostate cancer. They show Rb is an endogenous binding partner and inhibitor of BRD4 activity and Rb deficiency leads to high BRD4 activity and thus resistance to BET inhibitors. They further identify the downstream pathway GPCR-GNB1L-CREB as a functional module to confer resistance and inhibiting this pathway sensitizes BET inhibitors. The study is well performed with strict and high quality experiments and the findings are potentially important in both mechanistic insights and translation.

Below are my overall comments that can help to improve the study

Although Rb manipulation (KD or overexpression) has been shown to affect response to BET inhibitors, the BET inhibitor response in relation to the endogenous Rb should be shown. PC3, C-42 and DU145 represent cell lines with different Rb status and their IC50 of

BET inhibitors should be shown to support the role of endogenous Rb in this context. The study shows that Rb phosphorylation by CDK4/6 will reduce RB-BRD4 binding. Did the authors also check prostate cancer cells with wild-type RB but high CDK6/8 ? If so, one would expect that high CDK6/8 activity or amplification, like Rb deficiency, would also reduce Rb binding to BRD4, leading to resistance to BET inhibitors. Targeting CREB to sensitize BET inhibitors certainly sounds an interesting strategy to overcome BET inhibitor resistance in either Rb deficient or CDK6/8 high tumors. In the discussion, the author can expand the discussion about how often the Rb or CDK4/6 abnormalities, as well as CREB high expression, occur in PC and how these biomarkers can help with the translation of this finding.

Authors' Response to the Reviewers' Comments on NCOMMS-22-05599-T

We thank the Reviewers for the time they spent in evaluating our work and for their insightful comments, which we have considered thoroughly in generating the revised manuscript.

REVIEWER COMMENTS

Reviewer #1 (Remarks to the Author):

In this manuscript the authors show that RB1 mediates response to BET bromodomain inhibitors due to its direct interaction with BRD4. The authors mapped this interaction to the N-terminal part of RB1 and showed that RB1 phosphorylation by CDK4/6 kinase disrupts this binding. Based on ChIPseq for RB1 and BRD4 the authors determined that RB1 deletion enhances BRD4 chromatin binding to certain loci, and many of these encode genes involved in cAMP signaling. Subsequently they showed that inhibition of cAMP/CREB signaling restored BET inhibitor responsiveness of RB1 deficient cancer cells.

Specific points:

1. The experiments are done with a single cell line, PC3 cells. Although there are limited prostate cancer cell lines available, it would be important to reproduce the key findings in other lines.

Reply: We appreciate the insightful comments. As suggested, we repeated several key experiments in two more prostate cancer cell lines C4-2 and LNCaP. We found that RB knockdown also conferred resistance to BET inhibitors in both C4-2 and LNCaP cell lines (Figure 1b and Supplementary Figure 1a, c). Thus, using multiple cell lines we reproducibly demonstrated that RB deficiency causes BET inhibitor resistance in prostate cancer cells.

2. Most protein-interaction experiments are done with exogenously overexpressed tagged proteins. Validation by using endogenous proteins is important.

Reply: Great points. In addition to examining the interaction between endogenous RB and BET proteins in C4-2 cells (Figure 2a, c), we also performed similar experiments in PC-3 cells. We demonstrated that RB interacted with BET proteins at endogenous level in PC-3 cells (Figure 2b, d). Similar to the finding in C4-2 cells, the interaction between endogenous RB and endogenous BRD4, but not BRD2 and BRD3 was enhanced by λ protein phosphatase treatment of PC-3 cell lysate (Figure 2d). Thus, we demonstrated the interaction between endogenous RB and BET proteins in two different prostate cancer cell lines.

3. The methods section is very limited, many experiments are missing and even the ones listed are not described in sufficient detail to evaluate them.

Reply: We are very sorry that the methods for many experiments are missing. As suggested, we have provided very detailed information in the Methods section for all experiments.

4. The statistical tests are missing for all figures, the test used to calculate p-values have to be indicated for each figure.

Reply: We agreed. We have indicated the statistical tests used for p-value calculations in the figure legends of all relevant figures.

5. The quality of the BRD4 ChIPseq data seems questionable based on Figure 4. The authors should run QC in CISTROME to ensure the reliability of the data. Since many downstream experiments depend on this ChIPseq data, it is a key data in the manuscript.

Reply: We thank the Reviewer for the excellent suggestions. We agree that run QC in CISTROME is a great way to ensure the reliability of the data; however, we noticed that CISTROME platform has not been updated since 2014 and therefore this method was not pursued. Instead, we ran the comparison between our BRD4 ChIP-seq data with the data from ENCODE database. Our ChIP-seq data showed 34,809 BRD4 peaks in C4-2 prostate cancer cell line while ENCODE data reveal 14,705 BRD4 peaks in HepG2 liver cancer cell line. The occupancy differences of BRD4 in these two datasets could be due to, at least in part the use of different cell lines. Nevertheless, we found that there are 3,596 common peaks between these two datasets and the peak overlap is highly statistically significant ($p < 2.2e-16$). Importantly, two major BRD4 binding peaks in the *GNB1L* gene locus, the major focused target of many downstream studies, are present in both datasets despite the fact that different cell lines were used in these two studies (Review Figure 1). These data further confirm *GNB1L* as a putative BRD4 binding target gene. We would be happy to include this data in the manuscript if the Reviewer and Editor feel it is necessary.

Review Figure 1. UCSC Genome Browser screenshots showing BRD4 occupancy in the *GNB1L* gene locus in C4-2 cells (current study) and HepG2 cells (ENCODE).

6. The authors should provide an irrelevant chromatin-bound protein as control for the co-IP experiments, since looks like everything that's on the chromatin is detected in the RB co-IP in Figure 2.

Reply: Excellent point. We have included c-MYC, an irrelevant chromatin-bound protein as a negative control in our co-IP experiments shown in Figure 2a-d.

7. cAMP signaling is commonly associated with neural features and acquisition of neuroendocrine phenotype is a mechanisms of anti-androgen resistance in prostate cancer. Do the tumors with low RB and high *GNB1L* expression more likely to be resistant to anti-

androgens and have neuroendocrine features? Based on the association with survival (Fig 4o) this is a possibility.

Reply: These are excellent points. To address these questions, we knocked down RB in C4-2 (AR-positive) prostate cancer cell line. As shown in Supplementary Figure 6g, GNB1L protein was upregulated in RB knockdown (KD) cells compared to control cells, and GNB1L expression was largely depleted by co-knockdown of GNB1L. Similar to the findings in LNCaP/AR cell line as reported previously (Mu et al. Science 355 (6320): 84-88, 2017). RB KD alone not only decreased AR mRNA expression, but also induced resistance to the antiandrogen enzalutamide in C4-2 cells (Supplementary Figure 6i, j). However, co-knockdown of GNB1L failed to abolish RB KD-induced AR downregulation and enzalutamide resistance (Supplementary Figure 6i, j). Moreover, similar to the findings in LNCaP/AR cell line as reported previously (Mu et al. Science 355 (6320): 84-88, 2017), we found that RB KD alone induced upregulation of neuron-specific enolase (NSE), a neuroendocrine cell marker in C4-2 cells (Supplementary Figure 6i); however, this effect was not affected by co-knockdown of GNB1L either (Supplementary Figure 6i). Thus, these data support the previous report (Mu et al. Science 355 (6320): 84-88, 2017; Ku et al. Science 355 (6320): 78-83, 2017) that loss of RB enables antiandrogen resistance and neuroendocrine phenotype. However, our data suggest that none of these effects was mediated through RB loss-induced upregulation of GNB1L. Of course, our data cannot completely rule out the possibility that other cAMP related genes upregulated due to RB loss-induced aberrant activation of BRD4 may contribute to antiandrogen resistance and neuroendocrine phenotype in RB-deficient prostate cancer cells. We have included these new data and discussion in the revised manuscript.

8. The concentration of Palbociclib used (Figure 5a) is very high and the authors only see an affect at 5uM. This is not usual. Also the legend of Fig5a is shifted and the text is not above the lanes.

Reply: We agree that the concentration of Palbociclib used (Figure 5a) is high. As we demonstrated previously (Zhou et al. Cancer Research 81 (6): 1486-1499), relatively low concentrations (e.g. 1 μ M) of Palbociclib were used in estrogen receptor (ER)-positive breast cancer cell lines, and this effect could be attributed, at least in part to the fact that Cyclin D1 is a well-studied transcriptional target gene of ER. However, for some reasons, prostate cancer cells were not as sensitive to Palbociclib as ER-positive breast cancer cells and that is why high concentrations (e.g. 5 μ M) were used in our studies in prostate cancer cell lines.

According to the existing GNB1L protein WB data presented in Figure 5a, we agree that we did not see a drastic effect at the concentration lower than 5 μ M. We believe this was probably caused, at least in part by the fact that the GNB1L WB signal might be oversaturated. Indeed, as shown in the revised Figure 5a, the effect of Palbociclib at 1 μ M on GNB1L expression was effective when a shorter exposure (S.E.) WB film of GNB1L was used.

We are very sorry for the legend shift in this figure. The error has been corrected.

9. The cell proliferation growth assay is too short (Fig 6a), the authors should extend this until at least 3 independent points show statistically significant differences. right now only day 5 is

different, so they need to go up to day 10-12. It's also unclear how they have less than 1 day of treatment – probably the x-axis label is messed up?

Reply: We thank the Reviewer for the excellent point. We have repeated these experiments in both PC-3 and C4-2 cells. As shown in revised Figure 6a and Supplementary Figure 7h, we demonstrated that enhanced proliferation of RB-deficient PC-3 and C4-2 cells was statistically significantly inhibited by co-treatment with the BET inhibitor JQ1 and the CREB inhibitor 666-15. Given that we have observed statistically significant effects at three independent time points (Day 5, 6 and 7), we did not extend our experiments up to 10-12 days.

10. The graphs showing the cell proliferation and tumor volume differences are very busy and hard to differentiate the colors.

Reply: Per the request, we have revised the graphs by making the colors and illustrations much clearer.

11. Please do NOT use bar graphs, show the data as dot plots so each data point can be seen. This applies to Fig 6c, 5g, 6g, and many others.

Reply: We have changed all the bar graphs to dot plots as instructed.

Reviewer #2 (Remarks to the Author):

In this study, Ding et al explore the relationship between the RB and BRD4, specifically as it relates to resistance to BET inhibitor (BETi) drug treatment. The authors first created RB +/- prostate cancer cell lines which they tested against numerous clinically utilized drugs, finding that RB loss promoted resistance to BETi in an E2F1 independent manner. The authors further explored the relationship between RB and BRD4 through a number of binding studies, ultimately elucidating the interaction between the two proteins and the effect of RB phosphorylation on this interaction. Through BRD4 ChIP-Seq studies in the RB k/d lines, Ding et al nominated a number of putative target genes within the cAMP pathway for downstream characterization, ultimately showing GNB1L to be sensitive to changes in RB and BRD4 status. Finally, the authors show that the resistance to BETi seen in the RB k/d models can be overcome with combination treatment of the CREB inhibitor 666-15. This paper is well written and has a logical design. Additionally, the findings presented by the authors are noteworthy and have the potential to significantly influence prostate cancer clinical practice as well as future clinical trial opportunities in this space.

Comments:

1. The authors primarily utilize PC-3 and C42 cell lines for the RB +/- experiments. As recent studies have suggested HTS and CRPC cell lines have differential effects of RB loss (Mandigo et al 2021) it would be beneficial to see whether the same effects are seen in a HTS line such as LNCaP.

Reply: This is an excellent point. To address this concern, we performed experiments in hormone therapy sensitive cell line LNCaP, in which RB is intact. Similar to the results obtained

in RB intact CRPC cell lines, we found that RB knockdown also conferred resistance to BET inhibitors in HTS LNCaP cells (Supplementary Figure 1a, c). Thus, our data not only suggest that both HTS and CRPC cells can be resistant to BET inhibitors when RB becomes deficient, but also predict that CRPC tumors in patients could have higher chances to become BET inhibitor resistance due to the higher rate of RB loss in this tumor type in comparison to HTS tumors.

2. The experiments which evaluated the role for E2F1 in RB loss-induced resistance to JQ1 utilized AR-negative models. Can these experiments be recapitulated in the AR-positive C42 lines used earlier in the paper?

Reply: As suggested, we also evaluated the role of E2F1 in RB loss-induced resistance to JQ1 in AR-positive cell lines C4-2 and LNCaP. We demonstrated that similar results in AR-negative PC-3 cells (Figure 1g, h), depletion of E2F1 had no obvious effect on RB loss-induced BET inhibitor resistance in these two AR-positive prostate cancer cell lines (Supplementary Figure 2a-d).

3. What method was utilized to determine the differential binding for the ChIP-Seq studies?

Reply: We utilized DiffBind software from an open source (<https://bioconductor.org/packages/release/bioc/html/DiffBind.html>) to determine the differential binding for the ChIP-seq studies. This information has been included in the Method section in the revised manuscript.

4. The authors show that GNB1L expression patterns are similar in TCGA data, was this also true for the other genes in the cAMP pathway that were identified through the ChIP-Seq studies?

Reply: We performed similar analysis for other four genes in the cAMP pathway identified through the ChIP-seq studies. We demonstrated that two of these four genes were also significantly upregulated in RB-deficient PCa patient samples compared to RB-proficient samples (Supplementary Figure 5c-f). These data and those obtained from cell lines suggest that RB can regulate expression of these cAMP genes in cultured PCa cells and patient samples, but the regulation at certain gene loci could be influenced by the cellular contexts, especially in PCa samples from patients. The new data and the discussion have been included in the revised manuscript.

5. While the authors utilized RB/BRD4 alteration to toggle GNB1L expression, it would be interesting to how GNB1L overexpression alone affects the phenotypic results including resistance to BETi

Reply: We performed the experiments as suggested. We demonstrated that overexpression of GNB1L alone conferred resistance to BET inhibitors in both PC-3 and C4-2 cell lines (Supplementary Figure 6a-d). Importantly, we further showed that depletion of GNB1L abolished RB deficiency-induced BET inhibitor resistance in these two cell lines (Supplementary Figure 6e-h).

6. Can the authors comment on the potential drug-drug interaction between JQ1 and the CREBi compound used in the study?

Reply: Our data showed that co-treatment of JQ1 and CREBi resulted in greater anti-cancer effect than each individual compound both in vitro and in vivo (Fig. 6a-g). These data suggest that there was seemingly no obvious drug-drug interaction between these two compounds in both the in vitro and in vivo conditions. We have added this discussion in the revised manuscript.

Reviewer #3 (Remarks to the Author):

This study identifies a novel mechanism by which BET inhibitors develop resistance in prostate cancer. They show Rb is an endogenous binding partner and inhibitor of BRD4 activity and Rb deficiency leads to high BRD4 activity and thus resistance to BET inhibitors. They further identify the downstream pathway GPCR-GNB1L-CREB as a functional module to confer resistance and inhibiting this pathway sensitizes BET inhibitors. The study is well performed with strict and high quality experiments and the findings are potentially important in both mechanistic insights and translation.

Below are my overall comments that can help to improve the study

Although Rb manipulation (KD or overexpression) has been shown to affect response to BET inhibitors, the BET inhibitor response in relation to the endogenous Rb should be shown. PC3, C-42 and DU145 represent cell lines with different Rb status and their IC50 of BET inhibitors should be shown to support the role of endogenous Rb in this context.

Reply: These are excellent points. As instructed by the Reviewer, we examined the BET inhibitor response in relation to the endogenous Rb by measuring the IC50 of BET inhibitors in four different PCa cell lines with different RB status, including RB-deficient DU145 in which one allele of RB1 gene is deleted and the other has a small internal deletion mutation (Bookstein et al. Science 247 (4943): 712-715, 1990). Consistent with the status of total RB and functional RB inactivation (as indicated by S249/T252 phosphorylation), PC-3 and DU145 cell lines (in which either little/no RB protein or high level of phosphorylated RB was expressed) were much insensitive to BET inhibitors compared to C4-2 and LNCaP (in which low level of phosphorylated RB was expressed) (Supplementary Figure 7e-g). Notably, these results are consistent with the findings from an independent study reported previously that DU145 and PC-3 cells were much more insensitive to JQ1 compared to LNCaP cells (Asangani et al. Nature 510 (7504): 278-282, 2014).

The study shows that Rb phosphorylation by CDK4/6 will reduce RB-BRD4 binding. Did the authors also check prostate cancer cells with wild-type RB but high CDK6/8 ? If so, one would expect that high CDK6/8 activity or amplification, like Rb deficiency, would also reduce Rb binding to BRD4, leading to resistance to BET inhibitors.

Reply: Excellent point. To address this point, we overexpressed CDK4/Cyclin D1 in C4-2 and LNCaP cells which express a wild-type RB. We demonstrated that overexpression of

CDK4/Cyclin D1 not only increased RB phosphorylation (e.g. S249/T252-p) and GNB1L expression, but also caused resistance to BET inhibitors in both cell lines (Fig. 5h-k).

Targeting CREB to sensitize BET inhibitors certainly sounds an interesting strategy to overcome BET inhibitor resistance in either Rb deficient or CDK6/8 high tumors. In the discussion, the author can expand the discussion about how often the Rb or CDK4/6 abnormalities, as well as CREB high expression, occur in PC and how these biomarkers can help with the translation of this finding.

Reply: We thank the Reviewer for the excellent suggestions. We have included these points in the last paragraph in the Discussion section.

REVIEWER COMMENTS

Reviewer #1 (Remarks to the Author):

The authors have responded to each of the reviewers' specific points and revised the manuscript accordingly, but several responses regarding data quality and scientific rigor are incorrect or not sufficient.

The authors misunderstood the CISTROME question and did not address the ChIPseq data quality issue: the purpose of running the data in CISTROME is to run a QC pipeline to ensure the quality of the ChIP-seq data passes the required filters (Cistrome analysis pipeline link). Because the ChIPseq data is so critical for the conclusions of the paper and based on Figure 4 the quality is so poor, this needs to be addressed. Lastly, the CISTROME database has been updated regularly, the 2014 date is just when it was created. Looks like the authors did not even explore the website. But, even if they do not want to use CISTROME, please provide the following QC metrics:

Raw sequence median quality score
% Reads uniquely mapped
PCR bottleneck coefficient (PBC)
Number of merged Total/Fold 10/Fold 20 peaks
Fraction of reads in peaks (FRiP)
% Peaks in promoter/exon/intron/intergenic
% Top 5k peaks overlapping with union DHS

MYC is not really an irrelevant negative control for non-specific chromatin binding since it's in active transcriptional complexes that also contain BRD4 proteins. It's actually surprising that the authors did not get any signal for MYC in the RB pull down.

The fact that the authors had to use 5uM Palbociclib and did not see an effect below this makes the physiologic relevance of the findings questionable.

Reviewer #2 (Remarks to the Author):

The authors have done a satisfactory job making the required revisions based on the comments received. I believe this manuscript deserves publication.

Reviewer #3 (Remarks to the Author):

the authors have adequately addressed my c my comments by performing additional experiments. I have no further comments

Authors' Response to the Reviewers' Comments on NCOMMS-22-05599-A

We thank the Reviewers for the time they spent in evaluating our work and for their insightful comments, which we have considered thoroughly in generating the revised manuscript.

REVIEWER COMMENTS

Reviewer #1 (Remarks to the Author):

The authors have responded to each of the reviewers' specific points and revised the manuscript accordingly, but several responses regarding data quality and scientific rigor are incorrect or not sufficient.

The authors misunderstood the CISTROME question and did not address the ChIPseq data quality issue: the purpose of running the data in CISTROME is to run a QC pipeline to ensure the quality of the ChIP-seq data passes the required filters (Cistrome analysis pipeline link). Because the ChIPseq data is so critical for the conclusions of the paper and based on Figure 4 the quality is so poor, this needs to be addressed. Lastly, the CISTROME database has been updated regularly, the 2014 date is just when it was created. Looks like the authors did not even explore the website. But, even if they do not want to use CISTROME, please provide the following QC metrics:

Raw sequence median quality score
% Reads uniquely mapped
PCR bottleneck coefficient (PBC)
Number of merged Total/Fold 10/Fold 20 peaks
Fraction of reads in peaks (FRiP)
% Peaks in promoter/exon/intron/intergenic
% Top 5k peaks overlapping with union DHS

Reply: We thank the Reviewer for the excellent points. We therefore reached out Dr. Rongbin Zheng in the Department of Pediatrics at the Harvard Medical School in Boston who was one of the investigators helping with the development of the Cistrome pipeline.

Specifically, using the Cistrome pipeline, we compared our BRD4 ChIP-seq data obtained from both control and RB KD C4-2 cells to the BRD4 ChIP-seq results of 294 different human cell samples in the Cistrome database and demonstrated as follows:

1. Raw sequence median quality score: The scores of our four BRD4 ChIP-seq data are ranked at the top compared to the data of the 294 samples in the Cistrome database (Supplementary Fig. 5a and Supplementary Table 1).
2. % Reads uniquely mapped: The '% Reads uniquely mapped' of our four BRD4 ChIP-seq data are ranked at the average compared to the data of the 294 samples in the Cistrome database (Supplementary Fig. 5b and Supplementary Table 1).

3. PCR bottleneck coefficient (PBC): The PBCs of our four BRD4 ChIP-seq data are ranked at the average compared to the data of the 294 samples in the Cistrome database (Supplementary Fig. 5c and Supplementary Table 1).
4. Number of merged Total/Fold 10/Fold 20 peaks: For the total peaks and Fold 10 peaks, the number from our four BRD4 ChIP-seq data are ranked at the top 30% compared to the data of the 294 samples in the Cistrome database (Supplementary Fig. 5d,e and Supplementary Table 1).

For Fold 20 peaks, the numbers from our four BRD4 ChIP-seq data are ranked at the top compared to the data of the 294 samples in the Cistrome database (Supplementary Fig. 5f and Supplementary Table 1).

5. Fraction of reads in peaks (FRiP): The FRiPs of our four BRD4 ChIP-seq data are ranked at the average compared to the data of the 294 samples in the Cistrome database (Supplementary Fig. 5g and Supplementary Table 1).
6. % Peaks in promoter/exon/intron/intergenic: For ‘% Peaks in promoter’, the number of our four BRD4 ChIP-seq data are ranked at the top 20% compared to the data of the 294 samples in the Cistrome database (Supplementary Fig. 5h and Supplementary Table 1).

For ‘% Peaks in exon’, the number of our four BRD4 ChIP-seq data are ranked at the top 10% compared to the data of the 294 samples in the Cistrome database (Supplementary Fig. 5i and Supplementary Table 1).

For ‘% Peaks in intron’, the number of our four BRD4 ChIP-seq data are ranked in 30% compared to the data of the 294 samples in the Cistrome database (Supplementary Fig. 5j and Supplementary Table 1).

For ‘% Peaks in intergenic’, the number of our four BRD4 ChIP-seq data are ranked in 15% compared to the data of the 294 samples in the Cistrome database (Supplementary Fig. 5k and Supplementary Table 1).

These data suggest that BRD4 binding in C4-2 cells tended to be enriched more in the promoter and exon regions in the genome compared to the intronic and intergenic regions.

7. % Top 5k peaks overlapping with union DHS: the peaks of our four BRD4 ChIP-seq data are ranked in 30% compared to the data of the 294 samples in the Cistrome database (Supplementary Fig. 5l and Supplementary Table 1).

In summary, by running the Cistrome pipeline, we revealed that the quality of the BRD4 ChIP-seq data obtained from both control and RB KD C4-2 cells were comparable to the BRD4 ChIP-seq results of approximately 300 different human cell samples in the Cistrome database (Supplementary Fig. 5a-l and Supplementary Table 1).

MYC is not really an irrelevant negative control for non-specific chromatin binding since it's in active transcriptional complexes that also contain BRD4 proteins. It's actually surprising that the authors did not get any signal for MYC in the RB pull down.

Reply: Great point. Our data showed that RB binds to the BD1 domain and dampens BRD4 binding of acetylated histones on chromatin. Therefore, our interpretation of the MYC data is that RB-bound BRD4 proteins have limited ability to bind chromatin and hence they are likely in a protein complex different from MYC proteins that are engaged with chromatin (or chromatin-bound BRD4).

Nevertheless, we removed the MYC data and replaced with p50, one of the components of NF- κ B. We reported previously that RB binds to p65, another component of NF- κ B, but has no binding with p50 and inhibits NF- κ B transcriptional activity (Jin et al., Mol Cell, 73: 22-35, 2019). Consistent with this report, we found that RB did not bind p50 proteins; however, RB bound BRD4 in both C4-2 and PC-3 prostate cancer cell lines (Fig. 2a-d). A similar interpretation is that RB-bound BRD4 proteins have limited ability to bind chromatin and hence they are not in the same complex with NF- κ B proteins that are actively engaged with chromatin.

The fact that the authors had to use 5 μ M Palbociclib and did not see an effect below this makes the physiologic relevance of the findings questionable.

Reply: Excellent points. We repeated the experiments to determine whether palbociclib exerts its effect in a dose-dependent manner. We treated both PC-3 and C4-2 cells with different doses of palbociclib (0, 0.1, 0.5, 1, and 5 μ M). We found that palbociclib started to inhibit BRD4 binding in the GNB1L locus and GNB1L expression very obviously at the concentration of 0.5 μ M in both PC-3 and C4-2 cells (Fig. 5a-c and Supplementary Fig. 8e-g). Thus, our new data indicate that palbociclib enables to inhibit GNB1L expression at concentrations much lower than 5 μ M, suggesting that these findings are physiologically relevant.

Reviewer #2 (Remarks to the Author):

The authors have done a satisfactory job making the required revisions based on the comments received. I believe this manuscript deserves publication.

Reply: We thank the Reviewer for the positive comments on our current work.

Reviewer #3 (Remarks to the Author):

The authors have adequately addressed my comments by performing additional experiments. I have no further comments.

Reply: We thank the Reviewer for the positive comments on our current work.

REVIEWERS' COMMENTS

Reviewer #1 (Remarks to the Author):

The authors have addressed the remaining points, the revised manuscript is significantly improved.